



**Emissions of non-methane volatile organic compounds from combustion of domestic fuels**
**in Delhi, India**
Gareth J. Stewart[1], W. Joe F. Acton[2,a], Beth S. Nelson[1], Adam R. Vaughan[1], James R.
Hopkins[1,3], Rahul Arya[4,5], Arnab Mondal[4,5], Ritu Jangirh[4,5], Sakshi Ahlawat[4,5], Lokesh
Yadav[4,5], Sudhir K. Sharma[4,5], Rachel E. Dunmore[1], Siti S. M. Yunus[6], C. Nicholas Hewitt[2],
Eiko Nemitz[7], Neil Mullinger[7], Ranu Gadi[8], Lokesh. K. Sahu[9], Nidhi Tripathi[9], Andrew R.
Rickard[1,3], James D Lee[1,3], Tuhin K. Mandal[4,5] and Jacqueline F. Hamilton[1].
[1] Wolfson Atmospheric Chemistry Laboratories, Department of Chemistry, University of York, York, YO10 5DD, UK
[2] Lancaster Environment Centre, Lancaster University, Lancaster LA1 4YQ, UK
[3] National Centre for Atmospheric Science, University of York, York, YO10 5DD, UK
[4] CSIR-National Physical Laboratory, Dr. K.S. Krishnan Marg, New Delhi, Delhi 110012, India
[5] Academy of Scientific & Innovative Research, Ghaziabad, Uttar Pradesh- 201 002, India
[6] School of Water, Environment and Energy, Cranfield University, Cranfield, MK43 0AL, UK
[7] UK Centre for Ecology and Hydrology, Penicuik, EH26 0QB, UK
[8] Indira Gandhi Delhi Technical University for Women, Kashmiri Gate, New Delhi, Delhi 110006, India
[9] Physical Research Laboratory (PRL), Ahmedabad 380009, India
[a] Now at: School of Geography, Earth and Environmental Sciences, University of Birmingham, B15 2TT, Birmingham, UK
**Abstract**
29 different fuel types used in residential dwellings in northern India were collected from
across New Delhi (76 samples in total). Emission factors of a wide range of non-methane
volatile organic compounds (NMVOCs) (192 compounds in total) were measured during
controlled burning experiments using dual-channel gas chromatography with flame ionisation
detection (DC-GD-FID), two-dimensional gas chromatography (GC×GC-FID), proton-
transfer-reaction time-of-flight mass spectrometry (PTR-ToF-MS) and solid-phase extraction
two-dimensional gas chromatography with time-of-flight mass spectrometry (SPE-GC×GC-
ToF-MS). 94% quantification was achieved on average across all fuel types. The largest
contributors to emissions from most fuel types were small non-aromatic oxygenated species,
phenolics and furanics. The emission factors (in g kg$^{-1}$) for total gas-phase NMVOCs were:
fuel wood (18.7, 4.3-96.7), cow dung cake (62.0, 35.3-83.0), crop residue (37.9, 8.9-73.8),
charcoal (5.4, 2.4-7.9), sawdust (72.4, 28.6-115.5), municipal solid waste (87.3, 56.6-119.1)
and liquified petroleum gas (5.7, 1.9-9.8).
The emission factors measured in this study allow for better characterisation, evaluation and
understanding of the air quality impacts of residential solid fuel combustion in India.



## 1. Introduction

Biomass burning is the second largest source of trace gases to the troposphere, releasing around a half of global CO, ~ 20% of NO and ~ 8% of $CO_2$ emissions (Olivier et al., 2005; Wiedinmyer et al., 2011; Andreae, 2019). Biomass burning releases an estimated 400 Tg yr$^{-1}$ of non-methane volatile organic compounds (NMVOCs) annually (Akagi et al., 2011) and is the dominant source of both black carbon (BC) and primary organic aerosol (POA), representing 59% and 85% of global emissions respectively (Bond et al., 2013). Biomass burning includes open vegetation fires in forests, savannahs, agricultural burning and peatlands (Chen et al., 2017) as well as the biofuels used by approximately 3 billion people to meet their daily cooking and heating energy requirements worldwide (World Bank, 2017). A wide range of trace gases are released from biomass burning, in different amounts depending on the fuel type and the combustion conditions, meaning that detailed studies at the point of emission are required to accurately characterise emissions. The gases released lead to soil-nutrient redistribution (Ponette-Gonzalez et al., 2016; N'Dri et al., 2019), can themselves be toxic (Naeher et al., 2007) and can significantly degrade local, regional and global air quality through the photochemical formation of secondary pollutants such as ozone ($O_3$) (Pfister et al., 2008; Jaffe and Wigder, 2012) and secondary organic aerosol (SOA) (Alvarado et al., 2015; Kroll and Seinfeld, 2008). They can also lead to indoor air quality issues (Fullerton et al., 2008).

Emissions from biomass burning and their spatial distribution remain uncertain and estimates by satellite retrieval vary by over a factor of three (Andreae, 2019). Bottom-up approaches use information about emission factors and fuel usage. However, information for many developing countries, where solid fuel is a primary energy source, is particularly sparse. Toxic pollution from burning has been linked to chronic bronchitis (Akhtar et al., 2007; Moran-Mendoza et al., 2008), chronic obstructive pulmonary disease (Dennis et al., 1996; Orozco-Levi et al., 2006; Rinne et al., 2006; Ramirez-Venegas et al., 2006; Liu et al., 2007; PerezPadilla et al., 1996), lung cancer (Liu et al., 1993; Ko et al., 1997), childhood pneumonia (Smith et al., 2011), acute lower respiratory infections (Bautista et al., 2009; Mishra, 2003) and low birth weight of children (Boy et al., 2002; Yucra et al., 2011). Smoke from inefficient combustion of domestic solid fuels is the leading cause of conjunctivitis in developing countries (West et al., 2013). The harmful emissions from burning also resulted in an estimated 2.8-3.9 million premature deaths due to household air pollution (Kodros et al., 2018; WHO, 2018; Smith et al., 2014), of which 27% originated from pneumonia, 18% from strokes, 27% from ischaemic heart disease, 20% from chronic obstructive pulmonary disease and 8% from lung cancer, with hazardous



indoor air pollution responsible for 45% of pneumonia deaths in children less than 5 years old
(WHO, 2018). For this reason, hazardous indoor air pollution from the combustion of solid
fuels has been calculated to be the most important risk factor for the burden of disease in South
Asia from a range of 67 environmental and lifestyle risks (Lim et al., 2012; Smith et al., 2014).
The emissions from biomass burning fires are complex and can contain many hundreds to
thousands of chemical species (Crutzen et al., 1979; McDonald et al., 2000; Hays et al., 2002;
Hatch et al., 2018; Stewart et al., 2020a). Measurements of emissions by gas chromatography
(GC) have been made (Gilman et al., 2015; EPA, 2000; Wang et al., 2014; Stockwell et al.,
2016; Fleming et al., 2018; Tsai et al., 2003), as it has the potential to provide isomeric
speciation of emissions. However, it is of limited use in untargeted measurements from burning
due to the complexity of emissions, leading to large amounts of NMVOCs released not being
observed. Some of the main issues are that GC does not provide high time resolution
measurements and several instruments with different column configurations and detectors are
required to provide information on different chemical classes. Samples can also be collected
into canisters or sample bags and then analysed off-line (Sirithian et al., 2018; Wang et al.,
2014; Barabad et al., 2018), which can increase time resolution, but can also lead to artefacts
(Lerner et al., 2017).
Recent developments have allowed the application of proton-transfer-reaction mass
spectrometry (PTR-MS) to study the emissions from biomass burning (Warneke et al., 2011;
Yokelson et al., 2013; Brilli et al., 2014; Stockwell et al., 2015; Bruns et al., 2016; Koss et al.,
2018). PTR-MS uses proton transfer from the hydronium ion ($H_3O^+$) to ionise and
simultaneously detect most polar and unsaturated NMVOCs including aromatics, oxygenated
aromatics, alkenes, furanics and nitrogen containing volatile organic compounds (NVOCs) in
gas samples. PTR-MS can measure at fast acquisition rates of up to 10 Hz over a mass range
of 10 – 500 Th with very low detection limits of tens to hundreds of pptv (Yuan et al., 2016).
The more recently-developed technique of proton-transfer-reaction time-of-flight mass
spectrometry (PTR-ToF-MS) has allowed around 90% of emissions in terms of mixing ratio
from burning experiments to be quantified (Koss et al., 2018) and has also been used to study
the formation of SOA (Bruns et al., 2016). The main disadvantages of the PTR-ToF-MS
technique are its inability to speciate isomers/isobars, significant fragmentation of parent ions,
only being able to detect species with a proton affinity greater than water and the formation of
water clusters needing to be taken into account (Stockwell et al., 2015; Yuan et al., 2017). More
recently, measurements have also been made using iodide chemical ionisation time-of-flight





mass spectrometry (I⁻-CIMS), which is well suited to measuring acids and multifunctional oxygenates (Lee et al., 2014) as well as isocyanates, amides and organo-nitrate species released from biomass burning (Priestley et al., 2018). Multiple measurement techniques used in concert are therefore complementary, with the use of PTR-ToF-MS and simultaneous gas chromatography often alleviating some of the difficulties highlighted above.

Since the start of the century, rapid growth has resulted in India becoming the second largest contributor to NMVOC emissions in Asia (Kurokawa et al., 2013; Kurokawa and Ohara, 2019). However, effective understanding of the relative strength of different sources and subsequent mitigation has been limited by a deficiency of suitably detailed, spatially disaggregated emission inventories (Garaga et al., 2018). Current receptor-model studies have shown elevated NMVOC concentrations at an urban site in Delhi to be predominantly due vehicular emissions, with a smaller contribution from solid fuel combustion (Stewart et al., 2020b). However, approximately 60% of total NMVOC emissions from India in 2010 were shown to be due to solid fuel combustion (Sharma et al., 2015). A need has therefore been identified to measure local source profiles to allow evaluation with activity data to better understand the impact of unaccounted and unregulated local sources (Pant and Harrison, 2012).

Approximately 25% of worldwide residential solid fuel use takes place in India (World Bank, 2017), with approximately 25% of ambient particulate matter in South Asia attributed to cooking emissions (Chafe et al., 2014). Despite large government schemes, traditional solid fuel cookstoves remain popular in India because they are cheaper than ones that use liquified petroleum gas (LPG) and the meals cooked on them are perceived to be tastier (Mukhopadhyay et al., 2012). The total number of biofuel users has been sustained by an increasing population, despite the percentage use of biofuels decreasing as a proportion of overall fuel use due to increased LPG uptake (Pandey et al., 2014). Cow dung cakes remain prevalent as a fuel because they are cheap, readily available, sustainable and ease pressure on local fuel wood resources. Few studies have reported emissions data from cow dung cake (Stockwell et al., 2016; Koss et al., 2018; Fleming et al., 2018), leaving considerable uncertainty over the impact that cow dung cake combustion has on air quality. LPG usage has increased from around 100 to 500 million users over the same period, but only reflects around 10% of current rural fuel consumption (Gould and Urpelainen, 2018).

Inventories which include residential burning indicate a considerable emission source of around 6000-7000 kt yr⁻¹ (Pandey et al., 2014; Sharma et al., 2015). Burning is likely to have





a large impact on air quality in India, but considerable uncertainties exist over the total amount
of NMVOCs released owing to a lack of India specific emission factors and information related
to the spatial distribution of emissions.
Few studies exist measuring highly speciated NMVOC emission factors from fuels specific to
India. Recent studies using PTR-ToF-MS to develop emission factors, which are more
reflective of the range of species emitted from burning, have focussed largely on grasses, crop
residues and peat (Stockwell et al., 2015) as well as fuels characteristic of the western U.S.
(Koss et al., 2018). A previous study measured emission factors of NMVOCs from cow dung
cake using gas chromatography with flame ionisation detection (GC-FID) of 8-32 g kg$^{-1}$ (EPA,
2000). Fleming et al. (2018) quantified 76 NMVOCs from fuel wood and cow dung cake
combustion using *chulha* and *angithi* stoves by collecting samples into Kynar bags, transferring
their contents into canisters and off-line analysis using GC-FID, GC-ECD (electron capture
detector) and GC-MS. The emission factors measured from these 76 NMVOCs were 14 g kg$^{-1}$
for cow dung cake burnt in *chulha* stoves, 27 g kg$^{-1}$ for cow dung cake burnt in *angithi* stoves
and 6 g kg$^{-1}$ for fuel wood burnt in *angithi* stoves. An emission factor from one single dung
burn measured using PTR-MS was considerably larger at around 66 g kg$^{-1}$ (Koss et al., 2018).
Emissions from dung in Nepal have also been measured (Stockwell et al., 2016) by sampling
into whole air sample canisters followed by off-line analysis with GC-FID/ECD/MS and
Fourier-transform infrared spectroscopy (FTIR). However, very few speciated NMVOC
measurements were made and the emission factors were similar to those measured using just
GC (Fleming et al., 2018). Studies have also focussed on making detailed measurements, using
a range of techniques, from the burning of municipal solid waste (Christian et al., 2010;
Yokelson et al., 2011; Yokelson et al., 2013; Stockwell et al., 2015; Stockwell et al., 2016;
Sharma et al., 2019) and crop residues (Stockwell et al., 2015; Koss et al., 2018; Kumar et al.,

156    2018).

Detailed chemical characterisation of NMVOC emissions from fuel types widely used in the
developing world is much needed to resolve uncertainties in emission inventories used in
regional policy models and global chemical transport models. A greater understanding of the
key sources is required to characterise and hence understand air quality issues to allow the
development of effective mitigation strategies. In the present study we measure comprehensive
emission factors of NMVOCs from a range solid fuels characteristic to northern India.



## 2. Methods

### 2.1 Fuel collection and burning facility

A total of 76 fuels, reflecting the range of fuel types used in northern India, were collected from across New Delhi (see Figure 1 and Table 1). Cow dung cake usage was prominent in the north and west regions, whereas fuel wood use was more evenly spread across the state. Municipal solid waste was collected from Bhalaswa, Ghazipur and Okhla landfill sites. Collection also included less used local fuel types which were found being burnt, including crop residues, sawdust and charcoal. A low-cost LPG stove, widely promoted across India as a cleaner fuel (Singh et al., 2017), was also purchased to allow direct emission comparison with other local fuel types.

Fuels were burnt at the CSIR-National Physical Laboratory (NPL), New Delhi, under controlled conditions utilizing a combustion chamber that has been well described previously (Venkataraman et al., 2002; Saud et al., 2011; Saud et al., 2012; Singh et al., 2013), using expert local judgement to ensure conditions replicated real world burning conditions. Fuel (200 g) was rapidly heated to spontaneous ignition, with emissions convectively driven into a hood and up a flue to allow enough dilution, cooling and residence time to achieve the quenching typically observed in indoor environments. Samples were drawn down a sample line at 4.4 L min$^{-1}$ (Swagelok, ¼" PFA, < 2.2 s residence time) from the top of the flue, passed through a pre-conditioned quartz filter (ø = 47 mm, conditioned at 550 ℃ for 6 hours and changed between samples) held in a filter holder (Cole-Parmer, PFA) which was subsampled for analysis by PTR-ToF-MS, GC×GC-FID and DC-GC-FID instruments at a distance no greater than 5 m from the top of the flue.

Measurements of $n$-alkanes from $n$-tridecane ($C_{13}$) to eicosane ($C_{20}$) were also made from a subset of 29 burns using solid phase extraction disks (SPE, Resprep, $C_{18}$). Samples were passed through a cooling and dilution chamber designed to replicate the immediate condensational processes that occur in smoke particles approximately 5-20 mins after emission, yet prior to photochemistry which may change composition (Akagi et al., 2011). Further details of SPE sample collection are given in (Stewart et al., 2020a).





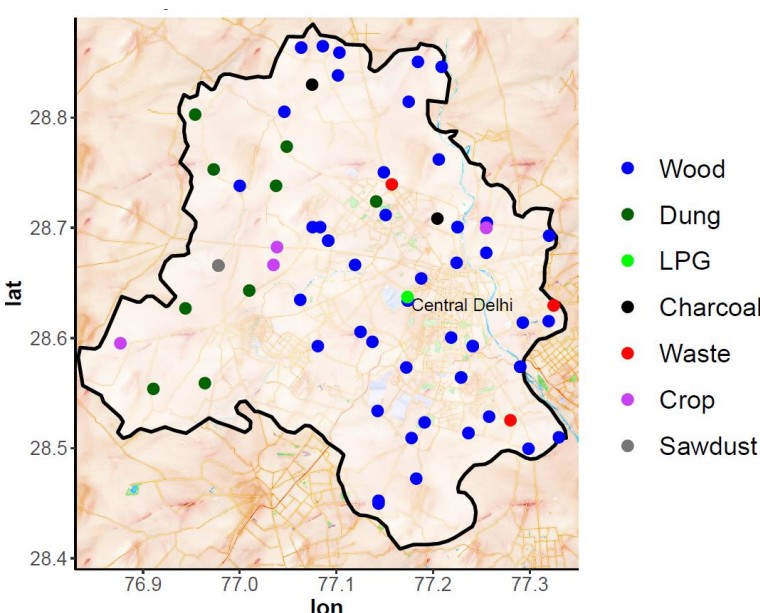

192

Figure 1. Locations across New Delhi used for the local surveys into fuel use and collection of representative biomass fuels. Map tiles by Stamen Design. Data by © OpenStreetMap contributors 2020. Distributed under a Creative Commons BY-SA License.

Table 1. Types and numbers of fuels burnt, the mean emission factor of total NMVOCs (TVOC) in g kg$^{-1}$ measured and standard deviation (SD) from all available burns. Discussion of TVOC calculation given is given in the text.

| Fuel woods | *n* | TVOC | SD | Other | *n* | TVOC | SD |
|---|---|---|---|---|---|---|---|
| *Azadirachta indica* | 3 | 18.6 | 7.9 | Cow dung cake | 8 | 61.9 | 18.4 |
| *Morus spp* | 4 | 27.4 | 21.1 | *Cocos nucifera* | 2 | 57.4 | 23.3 |
| *Melia azedarach* | 2 | 23.7 | 13.1 | Charcoal | 2 | 5.1 | 3.9 |
| *Shorea spp* | 2 | 9.8 | 2.2 | Sawdust | 2 | 71.3 | 60.8 |
| *Ficus religiosa* | 2 | 51.9 | 63.4 | Waste | 3 | 87.3 | 31.4 |
| *Syzygium spp* | 2 | 8.9 | 4.9 | LPG | 3 | 5.8 | 5.6 |
| *Ficus spp* | 2 | 7.1 | 1.2 | Cow dung cake mix | 1 | 34.7 | - |
| *Vachellia spp* | 2 | 13.5 | 9.7 | *Solanum melongena* | 2 | 13.6 | 6.5 |
| *Dalbergia sissoo* | 2 | 17.9 | 8.8 | *Brassica spp* | 2 | 41.0 | 45.5 |
| *Ricinus spp* | 2 | 8.5 | 2.5 | | | | |
| *Holoptelea spp* | 2 | 6.0 | 0.8 | | | | |
| Mixed woods | 1 | 6.1 | - | | | | |
| *Saraca indica* | 2 | 12.9 | 5.2 | | | | |
| *Populus spp* | 1 | 8.5 | - | | | | |
| *Pithecellobium spp* | 2 | 19.5 | 5.4 | | | | |
| *Eucalyptus spp* | 2 | 6.9 | 1.9 | | | | |
| *Prosopis spp* | 6 | 14.5 | 10.4 | | | | |
| *Mangifera indica* | 2 | 12.4 | 3.4 | | | | |
| Plywood | 8 | 26.6 | 24.3 | | | | |
| Processed wood | 2 | 33.7 | 17.2 | | | | |





### 2.2 PTR-ToF-MS

The PTR-ToF-MS (PTR 8000; Ionicon Analytik, Innsbruck) instrument from Physical Research Laboratory (PRL), Ahmedabad was used to quantify 107 masses and subsampled the common inlet line using ¼ inch PFA. Additional details of the PTR-ToF-MS system used in this study are given in previous papers (Sahu and Saxena, 2015; Sahu et al., 2016). The sample air was diluted into zero air, generated by passing ambient air (1 L min$^{-1}$) through a heated platinum filament at 550 °C, before entering the instrument with an inlet flow of 250 ml min$^{-1}$. Samples were diluted by either 5 or 6.25 times (50 ml min$^{-1}$ in 200 ml min$^{-1}$ zero air or 40 ml min$^{-1}$ in 210 ml min$^{-1}$ zero air). The instrument was operated with an electric field strength ($E/N$, where $N$ is the buffer gas density and $E$ is the electric field strength) of 120 Td. The drift tube temperature was 60 °C with a pressure of 2.3 mbar and 560 V applied across it.

Calibrations were performed twice a week using a gas calibration unit (Ionicon Analytik, Innsbruck). The calibration gas (Apel-Riemer Enironmental Inc., Miami) contained 18 compounds: methanol, acetonitrile, acetaldehyde, acetone, dimethyl sulphide, isoprene, methacrolein, methyl vinyl ketone, 2-butanol, benzene, toluene, 2-hexanone, *m*-xylene, heptanal, α-pinene, 3-octanone and 3-octanol at 1000 ppbv (± 5%) and β-caryophyllene at 500 ppbv (± 5%). This standard was dynamically diluted into zero air to provide a 6-point calibration. The normalised sensitivity (ncps/ppbv) was then determined for each mass using a transmission curve (Taipale et al., 2008). The maximum error in this calibration approach has been shown to be 21%. Peak assignment was assisted with results reported by previous burning studies and references therein (Brilli et al., 2014; Stockwell et al., 2015; Koss et al., 2018). The results may also contain other indistinguishable structural isomers not mentioned here.

Mass calibration and peak fitting of PTR-ToF-MS data were performed using PTRwid software (Holzinger, 2015). Count rates (cps) of each mass spectral peak were normalised to the primary ion ($H_3O^+$) and water cluster ($H_3O.H_2O)^+$ peaks, and mixing ratios were then determined for each mass using the normalised sensitivity. Where compounds known to fragment in the PTR-ToF-MS were identified, the mixing ratio of these species was calculated by summing parent ion and fragment ion mixing ratios. Before each burn, ambient air was sampled to provide a background for the measurement.

Petrol and diesel fuel samples were collected from an Indian Oil fuel station in Pusa, New Delhi, and the headspace analysed to allow comparison with benzene/toluene ratios. This was designed to analyse the ratios in evaporative emissions, as these have been shown to be an


important source of atmospheric NMVOCs (Srivastava et al., 2005; Rubin et al., 2006; Yamada
et al., 2015), which for example represented ~ 15% of anthropogenic UK NMVOC emissions
in 2018 (Lewis et al., 2020). Fuel samples were placed in a small metal container (¼" Swagelok
cap) which was connected to a two-way tap (¼" Swagelok). The tap was connected to a t-piece
(¼" Swagelok) which had a flow of zero air (250 ml min$^{-1}$) passed through it and could be
sampled by the PTR-ToF-MS. The tap was opened and closed which allowed the headspace of
fuels to be analysed.

## 2.3 DC-GC-FID

Gas chromatography was used to analyse entire burns to provide an integrated picture of
emissions from fuel types. The DC-GC-FID sampled 51 burns to measure 19 $C_2$-$C_7$ non-
methane hydrocarbons (NMHCs) and $C_2$-$C_5$ oxygenated VOCs (OVOCs) (Hopkins et al.,
2003). A 500 ml sample (1.5 L pre-purge of 100 ml min$^{-1}$ for 15 minutes, sample at 17 mL
min$^{-1}$ for 30 minutes) was collected (Markes International CIA Advantage), passed through a
glass finger at -30 °C to remove water and adsorbed onto a dual-bed sorbent trap (Markes
International ozone precursors trap) at -20 °C (Markes International Unity 2). The sample was
thermally desorbed (250 °C for 3 minutes) then split 50:50 and injected into two separate
columns for analysis of NMHCs (50 m × 0.53 mm $Al_2O_3$ PLOT) and OVOCs (10 m × 0.53
mm LOWOX with 50 μm restrictor to balance flow). The oven was held at 40 °C for 5 minutes,
then heated at 13 °C min$^{-1}$ to 110 °C, then finally at 8 °C min$^{-1}$ to 200 °C with a 30-minute hold.

## 2.4 GC×GC-FID

The GC×GC-FID was used to measure 58 $C_7$-$C_{12}$ hydrocarbons ($C_7$-$C_{12}$ alkanes, monoterpenes
and monoaromatics) and collected 3 L samples (100 ml min$^{-1}$ for 30 minutes) using an
adsorption-thermal desorption system (Markes International Unity 2). NMVOCs were trapped
onto a sorbent (Markes International U-T15ATA-2S) at -20 °C with water removed in a glass
cold finger at -30 °C, removed and heated to ~ 100 °C after each sample to prevent carryover
of unanalysed, polar interfering compounds. The sample was thermally desorbed (250 °C for 5
minutes) and injected splitless down a transfer line. Analytes were refocussed for 60 s using
liquid $CO_2$ at the head of a non-polar BPX5 held at 50 psi (SGE Analytical 15m × 0.15 μm ×
0.25 mm) which was connected to a polar BPX50 at 30 psi (SGE Analytical 2 m × 0.25 μm ×
0.25 mm) via. a modulator held at 180 °C (5 s modulation, Analytical Flow Products ELDV2-
MT). The oven was held for 2 minutes at 35 °C, then ramped at 2.5 °C min$^{-1}$ to 130 °C and held
for 1 minute with a final ramp of 10 °C min$^{-1}$ to 180 °C and hold of 8 minutes. The GC systems
were tested for breakthrough to ensure trapping of the most volatile components (see the



Supplementary Information S1 for an example from the GC×GC-FID). Calibration was carried
out using a 4 ppbv gas standard containing alkanes and aromatics (NPL UK) and through the
relative response of liquid standard injections to toluene for components not in this gas
standard, as detailed elsewhere (Dunmore et al., 2015; Stewart et al., 2020b). Integration of
peak areas was performed in Zoex GC image software (Zoex, USA). Peaks were individually
checked and where peaks were split in the software, they were manually joined. The areas
corresponding to alkane isomers were manually joined within the GC image software and
calibration performed by comparing the areas to the corresponding *n*-alkane. For both GC
instruments, blanks of ambient air were made at the beginning, middle and end of the day and
the mean subtracted from samples.

**274    2.5 GCxGC-ToF-MS**

Measurements were made of a subset of 29 burns of $C_{13}$-$C_{20}$ alkanes, as well as other gas-phase
species to assist with qualification of mases measured by PTR-ToF-MS, by adsorbing samples
to the surface of SPE disks with analysis by GC×GC-ToF-MS, as detailed in Stewart et al.
(2020a). Samples of 180 L were adsorbed to the surface of $C_{18}$ coated SPE disks (Resprep,
C18, 47 mm) prewashed with $2 \times 5$ mL acetone washes and $1 \times 5$ mL methanol wash. These
samples were collected at 6 L $min^{-1}$ over 30 minutes using a low volume sampler (Vayubodhan
Pvt.Ltd) which passed samples through a cooling and dilution chamber at 46.7 L $min^{-1}$.
Samples were then wrapped in foil, placed in an airtight bag and kept frozen until analysis.
SPE extracts were spiked with an internal standard (EPA 8270 Semivolatile Internal Standard
Mix, 2000 μg $mL^{-1}$ in DCM) and extracted using accelerated solvent extraction into ethyl
acetate. Extracts were analysed using GC×GC-ToF-MS (Leco Pegasus BT 4D) using a 10:1
split injection (1 μL injection, 4 mm taper focus liner, SHG 560302). The primary dimension
column was a RXI-5SilMS (Restek, 30 m × 0.25 μm × 0.25 mm) connected to a second column
of RXI-17SilMS (Restek, 0.25 μm × 0.25 mm, 0.17m primary GC oven, 0.1 m modulator, 1.42
m secondary oven, 0.31 m transfer line) under a He flow of 1.4 mL $min^{-1}$. The primary oven
was held at 40 °C for 1 min ramped at 3 °C $min^{-1}$ to 202 °C where it was held for 0.07 mins.
The secondary oven was held at 62 °C for 1 min then ramped at 3.2 °C $min^{-1}$ to 235 °C. The
inlet was held at 280 °C and the transfer line at 340 °C. A 5 s cryogenic modulation was used
with a 1.5 s hot pulse and 1 s cool time between stages.
Peaks assignment was conducted through comparison of retention times to known standards
and comparison to the National Institute of Standards and Technology (NIST) mass spectral





library. Peaks with no standard available were tentatively identified if the NIST library hit
exceeded 700 (Stein, 2011).
Integration was carried out within the ChromaTOF 5.0 software package (Leko, 2019). Eight
blank measurements were made at the beginning and end of the day by passing air from the
chamber (6 L min$^{-1}$ for 30 mins) through the filter holder containing a PTFE filter and an SPE
disk. Blank corrections have been applied by subtracting the mean of blank values closest to
measurement of the sample. An 8-point calibration was performed for $n$-alkanes using a
commercial standard ($C_7$-$C_{40}$ saturated alkane standard, certified 1000 µg mL$^{-1}$ in hexane,
Sigma Aldrich 49452-U) diluted in the range 0.25 – 10 µg ml$^{-1}$.
**3. Results**
**3.1 Chromatography**
Figure 2 shows GC×GC-FID chromatograms obtained from collecting the emissions during
the combustion of LPG (Figure 2A), *Saraca indica* fuel wood (Figure 2B), cow dung cake
(Figure 2C) and municipal solid waste (Figure 2D). Figure 2D is labelled to show the position
of NMVOCs measured and displays a homologous series of $n$-alkanes from $n$-heptane ($C_7$) to
$n$-tetradecane ($C_{14}$) along the bottom, with the 1-alkenes positioned to the left. Above are more
polar species such as monoterpenes, aromatics from benzene to substituted monoaromatics
with up to 5 carbon substituents, and at a higher second dimension retention time even more
polar species, such as styrene.
Many peaks were present in the chromatograms for cow dung cake and municipal solid waste,
and these fuels released significantly more NMVOCs per unit mass than fuel wood and LPG
(see Table 1). Cow dung cake and municipal solid waste released a range of NMVOCs
including $n$-alkanes, alkenes, and aromatics. The municipal solid waste (Figure 2D) showed a
particularly large and tailing peak 22 owing to large emissions of styrene. Several unidentified
peaks were observed in these complex samples which were broad in the second dimension.
These were assumed to be from polar, oxygenated species formed during burning such as
phenol. These species could not be identified and were not analysed using the GC×GC-FID.
Peaks have been omitted if these species were found to interfere significantly. Analysis has
only been carried out using the DC-GC-FID from ethane ($C_2$) to $n$-hexane ($C_6$) owing to the
significant presence of coeluting peaks. The large peak in the LPG chromatogram (Figure 2, 1º
~6 min, 2º ~ 0.5 s) was from unresolved propane and butane because of the high concentrations
from this fuel source.

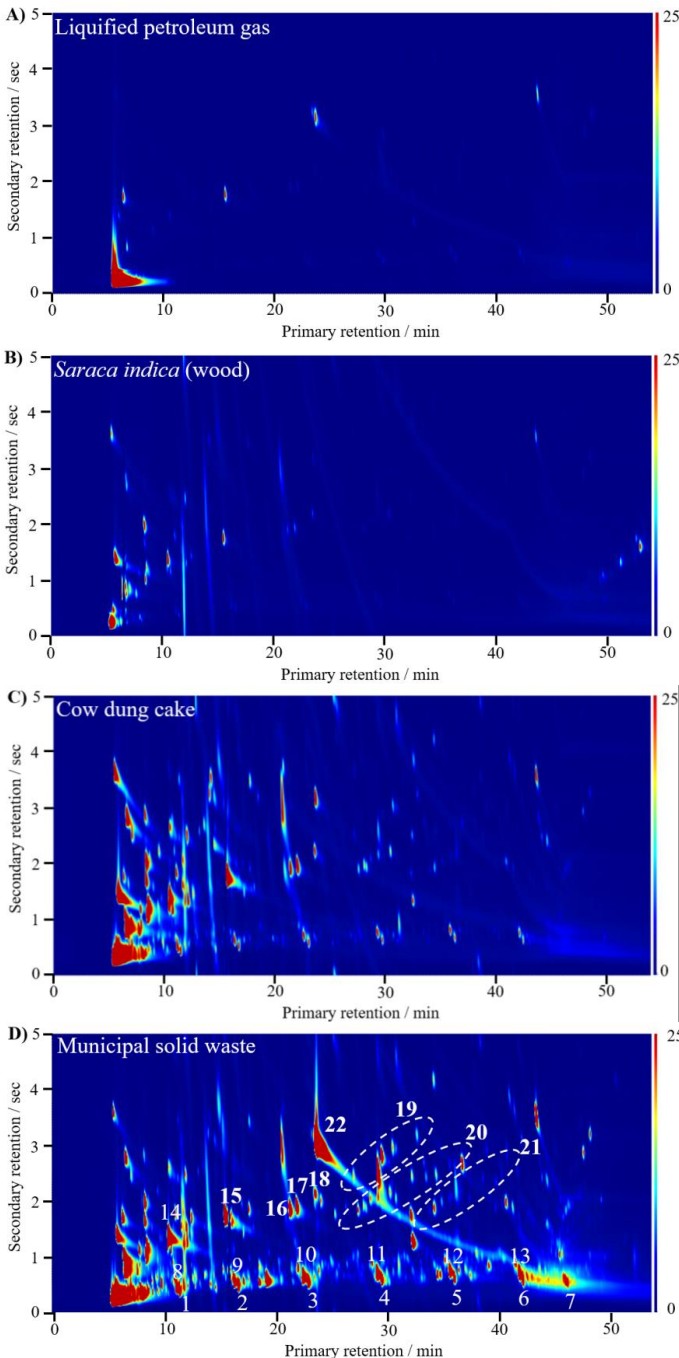

Figure 2. GC×GC-FID chromatograms from burning (A) = LPG, (B) = *Saraca indica* (fuel wood), (C) = cow dung cake and (D) = municipal solid waste samples where 1-7 = *n*-octane – *n*-tetradecane, 8-13 1-octadecene – 1-tridecene, 14 = benzene, 15 = toluene, 16 = ethylbenzene, 17 = *m/p*-xylene, 18 = *o*-xylene, 19 = $C_3$ substituted monoaromatics, 20 = $C_4$ substituted monoaromatics, 21 = $C_5$ substituted monoaromatics and 22 = styrene.





**3.2 PTR-ToF-MS**

Figure 3 shows an example concentration-time series measured by the PTR-ToF-MS for a cow dung cake burn. A sharp rise in NMVOC emissions was seen from the start of the burn which decreased as the fuel was combusted. Emissions of small oxygenated species as well as phenolics and furanics were dominant throughout most of the burn. At the beginning, a greater proportion of lower mass species were released, as shown in the binned mass spectrum of regions A/B in Figure 3. At the end in the smouldering phase, emissions were dominated by heavier and lower volatility species (Figure 3, Region C). A previous study indicated larger molecular weight phenolics were from low temperature pyrolysis (Sekimoto et al., 2018).

Figure 4 shows the cumulative mass of species measured from burns of fuel wood, cow dung cake, municipal solid waste and charcoal as a proportion of the total mass of NMVOCs quantified using PTR-ToF-MS. The results were similar to those reported by Brilli et al. (2014) and Koss et al. (2018): 65-90% of the mass of NMVOCs at emission originated from around 40 NMVOCs, with around 70-90% identification by mass when quantifying around 100 NMVOCs. The largest contributors to the NMVOC mass from burning of fuel wood and cow dung cake were methanol (*m/z* 33.034), acetic acid (*m/z* 61.028) and a peak that reflected the sum of hydroxyacetone, methyl acetate and ethyl formate (*m/z* 75.043). For municipal solid waste samples around 28% of total mass was from methyl methacrylate (*m/z* 101.059) and styrene (*m/z* 105.068), and two of the three municipal solid waste samples released significant quantities of styrene, most likely the result of degradation of polystyrene in the samples.

Figure 5 shows a time series for phenolics and furanics from the burning of an example fuel wood. Most species of similar functionality tracked each other. Stockwell et al. (2015) demonstrated that benzene, phenol and furan could act as tracers for aromatic, phenolic and furanic species released from biomass burning. Figure 5A shows that heavier, more substituted phenolics appeared to be released at cooler temperatures. Guaiacol (dark blue) was released at the start of the flaming phase before the temperature increased and more phenol (red) was released at higher burn temperatures. Later in the burn, a larger proportion of vinyl guaiacol (pink) and syringol (yellow) were emitted. This agreed well with previous results which showed that species emitted from lower temperature depolymerisation had a larger proportion of low-volatility compounds compared to higher temperature processes during flaming (Sekimoto et al., 2018; Koss et al., 2018). Figure 5B shows timeseries of furanic species, with most species showing similar characteristics throughout the burn. The only species to peak later in the burn was 2-hydroxymethyl-2-furan.



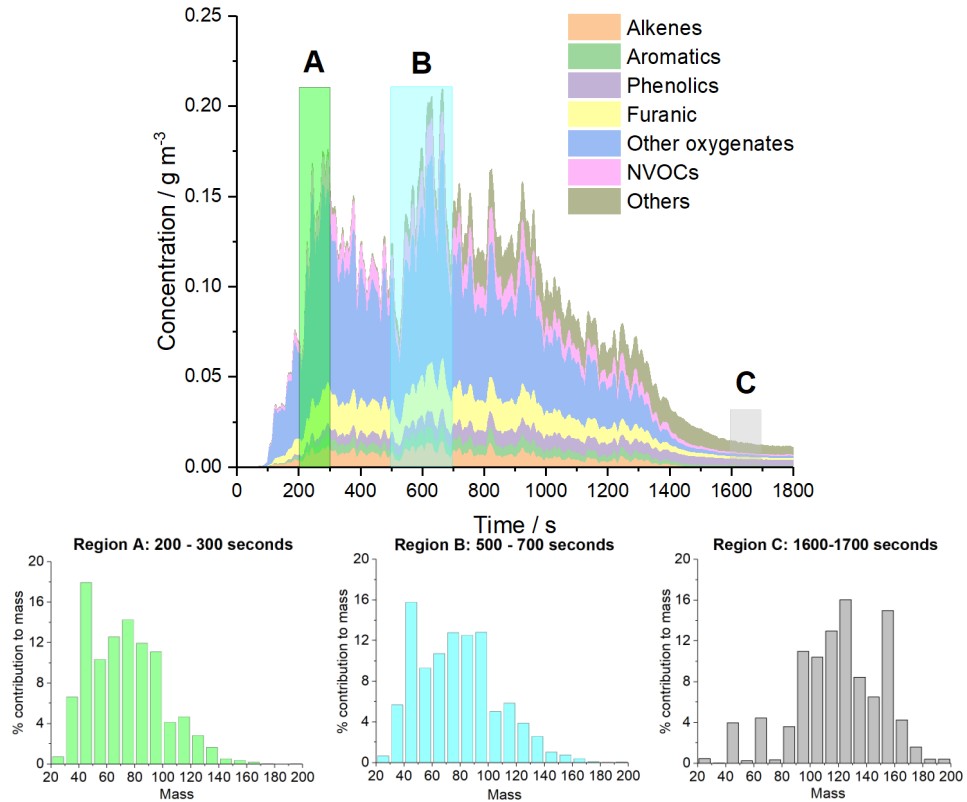

Figure 3. PTR-ToF-MS concentration-time series during the first 30 minutes of a cow dung cake burn coloured by functionality with regions A, B and C displaying mass spectra placed into $m/z$ bins of 10 Th. Fuel collected from Pitam Pura, New Delhi.

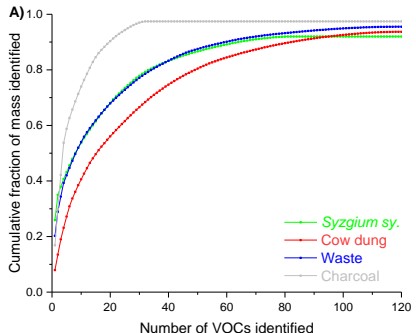
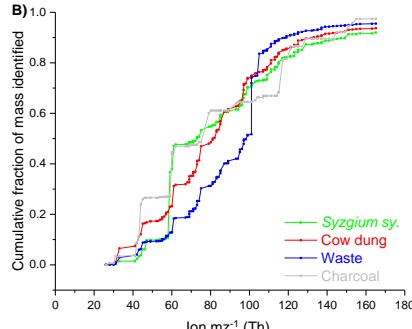

Figure 4. Cumulative NMVOC mass identified from PTR-ToF-MS compared with total NMVOC signal from PTR-ToF-MS with (A) ordered by decreasing NMVOC mass contribution and (B) ordered by ion mass. High quantification of emissions from charcoal was due to a low emission factor (2.4 g kg$^{-1}$).





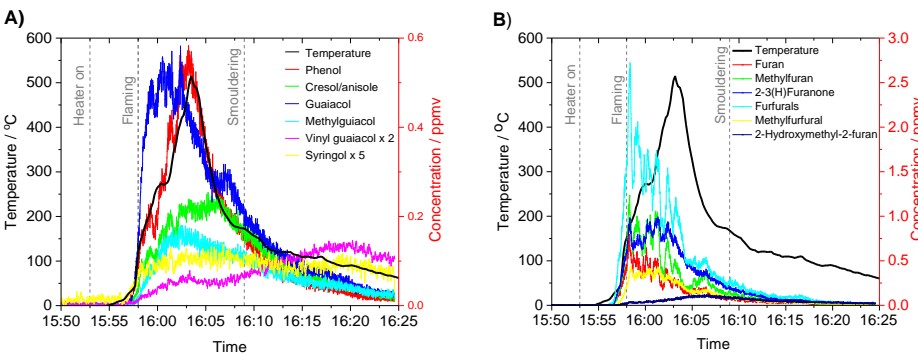

Figure 5. Timeseries analysis of phenolic and furanic compounds released from burning of *Azadirachta indica* which released 27.0 g kg$^{-1}$ of NMVOCs. Temperature corresponds to the increase in temperature above ambient measured in the flame directly above the combustion experiment.

### 3.3 Comparison of emissions data obtained with different instruments

Previous instrument inter-comparisons from biomass burning samples were between PTR-MS, GC-MS and open path FTIR (Gilman et al., 2015) and between PTR-ToF-MS, FTIR, broadband cavity-enhanced spectroscopy (ACES) and I$^-$-CIMS (Koss et al., 2018). Gilman et al. (2015) showed generally good agreement of slopes of measured emission factors between benzene, ethyne, furan, ethene, propene, methanol, toluene, isoprene and acetonitrile using different instruments/techniques with slopes of ~ 1 ± 30% and correlation coefficients > 0.9. Koss et al. (2018) showed mean measured values of most NMVOCs from all burns with other instruments compared to the PTR-ToF-MS which agreed within a factor of two and had correlation coefficients > 0.8 for most species except butadienes, furan, hydroxyacetone, furfural, phenol and glyoxal. These previous comparisons underline the challenges faced with quantitative NMVOC measurements from burning experiments. Figure 6 shows a comparison of measurements made using the DC-GC-FID, GC×GC-FID and PTR-ToF-MS techniques. Bar plots show that the mean and lower/upper quartiles of all measurements agreed within a factor of two. The correlation coefficient between different instruments is given in blue circles, with all > 0.8. Generally, the mean values measured for the PTR-ToF-MS were slightly larger than using the GC instruments, which was attributed to the presence of other undistinguishable structural isomers measured by the PTR-ToF-MS. Comparison between DC-GC-FID and GC×GC-FID measurements were also complicated by high levels of coelution of additional NMVOC species released from combustion with similar retention times ($R_t$) to benzene/toluene ($R_t$ = 21/25 mins) on the DC-GC-FID instrument. Generally, the smallest values were measured with the GC×GC-FID instrument, consistent with the greatest ability to speciate isomers and





limit the impacts of coelution. Significant efforts were made to synchronise the sample periods
for the three instruments as best as possible; however, slight uncertainty existed over the exact
time each instrument started measuring when calculating mean sample windows ($\pm 30\,\mathrm{s}$). These
factors combined, may help to explain the slight differences observed between different
instruments during this study. When multiple instruments have measured the same NMVOC
in this study, preference was given to the data from the GC×GC-FID due to the ability of this
instrument to resolve coeluting peaks, followed by the DC-GC-FID and then the PTR-ToF-
MS.

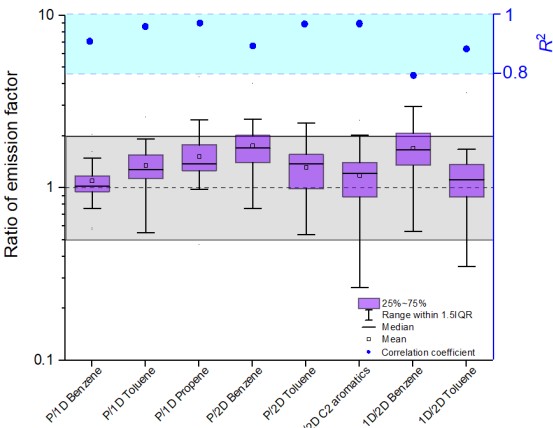


Figure 6. Comparison of PTR-ToF-MS to DC-GC-FID and GC×GC-FID with the black dashed line
representing slopes equal to one, grey shaded region = slopes agreeing within a factor of two, shaded
blue region indicating correlation coefficients > 0.8 and P = PTR-ToF-MS, 1D = DC-GC-FID and 2D
= GC×GC-FID.

**3.4 NMVOC emission factors from biomass fuels**

Figure 7 shows a detailed breakdown of the mean NMVOC emission factors by fuel type
measured for all 76 burns (see the Supplementary Information S2 for values). The data is split
by functionality to show trends for different chemical types. This shows that burning released
a large amount of different NMVOCs, across a wide range of functionalities, molecular
weights, and volatilities. The large variety of NMVOCs are likely to have different influences
on $O_3$ formation, SOA production and the toxicity of emissions.
Figure 7A shows very large emissions of smaller oxygenated species which were driven by
methanol, acetic acid and the unresolved combined peak for hydroxy acetone, methyl acetate
and ethyl formate. For the fuel wood samples, acetic acid/glycolaldehyde (2.6 g kg$^{-1}$), methanol
(1.8 g kg$^{-1}$) and acetaldehyde (0.6 g kg$^{-1}$) compared well with mean values reported by Koss et



al. (2018) for pines, firs and spruces (2.7/1.3/1.2 g kg$^{-1}$) and the mean values measured by
Stockwell et al. (2015) mainly from crop residues, grasses and spruces (1.6/1.3/0.9 g kg$^{-1}$). The
emission factor from this study for the unresolved peak of hydroxy acetone, methyl acetate and
ethyl formate (1.4 g kg$^{-1}$) was larger than those previously reported by Koss et al. (2018) and
Stockwell et al. (2015) of 0.55 and 0.25 g kg$^{-1}$, respectively.
Figure 7B shows that there were large emissions of furans and furanones from combustion,
mainly from methyl furans, furfurals, 2-(3H)-furanone, methyl furfurals and 2-methanol
furanone. The World Health Organisation consider furan a carcinogenic species of high-
priority (WHO, 2016) with furan and substituted furans, suspected to be toxic and mutagenic
(Ravindranath et al., 1984; Peterson, 2006; Monien et al., 2011). Furan emissions originate
from the low temperature depolymerisation of hemi-cellulose (Sekimoto et al., 2018) and from
large alcohols and enols in high-temperature regions of hydrocarbon flames (Johansson et al.,
2016). The OH chemistry of furans has been the subject of several studies (Bierbach et al.,
1994; Bierbach et al., 1995; Tapia et al., 2011; Liljegren and Stevens, 2013; Strollo and
Ziemann, 2013; Zhao and Wang, 2017; Coggon et al., 2019) and often produces more reactive
products such as butenedial, 4-oxo-2-pentenal and 2-methylbutenedial (Bierbach et al., 1994;
Gómez Alvarez et al., 2009; Aschmann et al., 2011, 2014). Photo-oxidation of furans may also
be a potentially important source of small organic acids such as formic acid (Wang et al., 2020).
Oxidation can also occur by nitrate (Berndt et al., 1997; Colmenar et al., 2012) or chlorine
radicals (Cabañas et al., 2005; Villanueva et al., 2007). As a result, furans have recently been
shown to be some of the species with highest OH reactivity from biomass burning, causing an
estimated 10% of the O$_3$ produced by the combustion emissions in the first 4 hours after
emission (Hartikainen et al., 2018; Coggon et al., 2019). Oxidation of furans can lead to SOA
production (Gómez Alvarez et al., 2009; Strollo and Ziemann, 2013) with an estimated 8-15%
of SOA caused by furans emitted by burning of black spruce, cut grass, Indonesian peat and
ponderosa pine and 28-50% of SOA from rice straw and wiregrass (Hatch et al., 2015),
although SOA yields are still uncertain for many species (Hatch et al., 2017).
Phenols are formed from the low-temperature depolymerisation of lignin (Simoneit et al., 1993;
Sekimoto et al., 2018) which is a polymer of randomly linked, amorphous high-molecular
weight phenolic compounds (Shafizadeh, 1982). Owing to their high emission ratios and SOA
formation potentials, phenolic compounds contribute significantly to SOA production from
biomass-burning emissions (Yee et al., 2013; Lauraguais et al., 2014; Gilman et al., 2015;
Finewax et al., 2018). Figure 7C shows that the largest phenolic emissions from fuel wood in



this study were methoxyphenols, with significant contributions from phenol, guaiacol, cresols
and anisole. Phenolic emissions from sawdust were dominated by guaiacol and creosol.
Phenolic emissions from coconut shell were greatest, most likely as a result of the lignin rich
nature of coconut shell (Pandharipande, 2018). The larger mean emission of furanics (3.2 g kg$^{-1}$
$^{1}$) compared to phenolics (1.1 g kg$^{-1}$) from fuel wood was consistent with wood being composed
of around 75% cellulose/hemicellulose and 25% lignin (Sjöström, 1993).
Figure 7D shows that the largest alkene emission was styrene from burning municipal solid
waste, likely caused by the presence of polystyrene in the fuel. Emissions of alkenes from fuel
woods were dominated by ethene and propene, species with high photochemical ozone creation
potential (Cheng et al., 2010). Monoterpenes¸ which are extremely reactive with the OH radical
(Atkinson and Arey, 2003), were emitted from combustion of sawdust, cow dung cake and
municipal solid waste samples.
Ethane and propane dominated the alkane emissions for fuel wood samples (see Figure 7E). A
wider range of alkanes from $C_2$-$C_{20}$ were observed from combustion of coconut, cow dung
cake and municipal solid waste. The largest alkane emission by mass was from LPG due to
unburnt propane and butane.
Nitrogen containing VOCs (NVOCs) are formed from the volatilisation and decomposition of
nitrogen-containing compounds within the fuel, mainly from free amino acids but can also be
from pyrroline, pyridine and chlorophyll (Leppalahti and Koljonen, 1995; Burling et al., 2010;
Ren and Zhao, 2015). NVOCs are of interest because nitrogen may be important in the
development of new particles (Smith et al., 2008; Kirkby et al., 2011; Yu and Luo, 2014) which
act as cloud condensation nuclei (Kerminen et al., 2005; Laaksonen et al., 2005; Sotiropoulou
et al., 2006) and alter the hydrological cycle by forming new clouds and precipitation (Novakov
and Penner, 1993). They can also contribute to light-absorbing brown carbon (BrC) aerosol
formation, effecting climate (Laskin et al., 2015). Additionally, NVOCs can be extremely toxic
(Ramírez et al., 2012, 2014; Farren et al., 2015). Cow dung cake was the largest emitter of
NVOCs (4.9 g kg$^{-1}$), releasing large amounts of acetonitrile and nitriles, likely to have a large
impact on the toxicity and chemistry of emissions (see Figure 7F).





Figure 7. Measured emission factors grouped by functionality.





Figure 7G shows emissions of aromatics from fuel wood, cow dung cake and municipal solid
waste were principally benzene, toluene and naphthalenes. Large emissions of benzene were
unsurprising as biomass burning is the largest global benzene source (Andreae and Merlet,
2001). Emissions of benzene, toluene, ethylbenzene and xylenes (BTEX) from cow dung cake
(0.5-1.7 g kg$^{-1}$) were in line with previous measurements of 1.3 g kg$^{-1}$ (Koss et al., 2018) and
1.8 g kg$^{-1}$ (Fleming et al., 2018) but lower than the 4.5 g kg$^{-1}$ reported from cow dung cake
combusted from Nepal (Stockwell et al., 2016). Emissions of BTEX from municipal solid
waste burning (0.9– 2.6 g kg$^{-1}$) were comparable to that measured previously (3.5 g kg$^{-1}$)
(Stockwell et al., 2016).
Figure 7H shows a qualitative comparison of species such as ammonia, HCN and dimethyl
sulphide which were measured during experiments, but could not be accurately quantified as
their sensitivity was too different from the NMVOCs used to build the transmission curve. Cow
dung cake emitted significantly more of these species than other fuel types.
Table 2 shows the total emission factors of NMVOCs for different fuel types. These have been
determined by calculating the total volume of air convectively drawn up the flue and relating
this to the mass of fuel burnt (see the Supplementary Information S3 for details). Emission
factors have been calculated over a 30-minute period, in line with the GC sample time, with
any small emissions after this sample window not included. The total emission factor has been
calculated as the sum of the PTR-ToF-MS signal, excluding reagent ion peaks (< $m/z$ 31 Th)
water cluster peaks ($m/z$ 37 Th) and isotope peaks identified for all masses (SIS, 2016). The
emission factors for all alkanes measured were also included as alkanes up to $n$-hexane had
proton affinities less than water and larger alkanes had proton affinities similar to water (Ellis
and Mayhew, 2014; Wróblewski et al., 2006). This low sensitivity meant that no peaks were
present in the PTR-ToF-MS spectra for these larger species. Further information on the
calculation of the total emission factor is given in the Supplementary Information S4.
Table 2. Mean total NMVOC emission factors (g kg$^{-1}$, including IVOC fraction) where high/low EF
represent the largest/smallest emission factor measured for a given sample type (g kg$^{-1}$) and IVOC is
the sum of emission factors of species with a mass greater than benzaldehyde (g kg$^{-1}$) where $n$ = number
of measurements made.

|  | Wood | Dung | Waste | LPG | Charcoal | Sawdust | Crop |
|---|---|---|---|---|---|---|---|
| NMVOC | 18.7 | 62.0 | 87.3 | 5.7 | 5.4 | 72.4 | 37.9 |
| High EF | 96.7 | 83.0 | 119.1 | 9.8 | 7.9 | 114.0 | 73.8 |
| Low EF | 4.3 | 35.3 | 56.3 | 1.9 | 2.4 | 28.3 | 8.9 |
| IVOC | 3.5 | 12.6 | 13.2 | 0.2 | 1.4 | 16.9 | 8.0 |
| $n$ | 51 | 8 | 3 | 3 | 2 | 2 | 6 |



Coconut shell, sawdust, cow dung cake and municipal solid waste released the greatest mass
of NMVOC per kg of fuel burnt. The mean emission factor for all fuel woods (18.7 g kg⁻¹) was
comparable to that for chaparral (16.6 g kg⁻¹) measured using PTR-ToF-MS by Stockwell et
al. (2015). This may be due to similarities between north Indian fuel wood types with chaparral,
which is characterised by hot dry summers, and mild wet winters. The mean fuel wood
emission factor was smaller than Stockwell et al. (2015) reported for coniferous canopy (31.0
g kg⁻¹). The NMVOC emission measured for cow dung cake (62.0 g kg⁻¹) was comparable to
that previously reported (66.3 g kg⁻¹) in literature using PTR-ToF-MS (Koss et al., 2018), but
2-3 times larger than that measured by GC-FID/ECD/MS likely due to those techniques
missing significant amounts of emissions (Fleming et al., 2018). Whilst the total emissions
reported by Fleming et al. (2018) might therefore be an underestimate, it is noteworthy that the
emission factors measured by Fleming et al. (2018) in *angithi* stoves for cow dung cake were
~ factor of 4 greater than fuel wood under the same conditions. This result was comparable to
this study which showed that cow dung cake emissions were ~ factor of 3 larger than fuel wood,
however the techniques used here targeted a greater proportion of total emissions. Moreover,
Fleming et al. (2018) reported emission factors from combustion of biomass fuels from a
neighbouring state, Haryana, and there may be slight heterogeneity between the different fuels
collected in both studies. NMVOC emissions from municipal solid waste (87.3 g kg⁻¹) were
significantly larger than the 7.1 g kg⁻¹ (Stockwell et al., 2015) and 33.8 g kg⁻¹ (Stockwell et al.,
2016) previously reported. This was likely due to differences in composition and moisture
content of the fuels collected from Indian landfill sites for the present study, compared with the
daily mixed waste and plastic bags collected at the US fire services laboratory (Stockwell et
al., 2015) and a variety of mixed waste and plastics collected from around Nepal (Stockwell et
al., 2016). It seems noteworthy that combustion experiments of fuels collected from developing
countries in Stockwell et al. (2016) had larger emission factors than those collected from, and
burnt at a laboratory (Stockwell et al., 2015). The mean crop residue combustion emission
factor (37.9 g kg⁻¹) was comparable to that reported by Stockwell et al. (2015) (36.8 g kg⁻¹),
despite the small number of samples in this study and compositional differences.
Table 2 also shows an approximation for the mean amount of IVOCs released by fuel type.
The IVOC fraction has been approximated by considering all NMVOCs with a mass greater
than benzaldehyde to be IVOCs. This approach was approximate as vapour pressures depend
on both mass and functionality. The fuels tested in this study showed that mean emissions of
IVOC species represented approximately 18 – 27% of total emissions from all fuel types other





than LPG. This agreed well with the IVOC fraction reported by Stockwell et al. (2015) of ~
14-26%. This demonstrated that biomass burning is potentially a large global source of IVOCs.
Further studies are required to better understand the contribution of IVOC emissions from
biomass burning to SOA formation.
Figure 8A shows the mean total emissions measured in this study for different fuel types split
by functionality. Large variability in total emissions were observed for fuel woods, with
emission factors from individual burns varying by ~ factor 20. Figure 8B shows the mean
emissions by functionality as a proportion of total emissions averaged by overall fuel type.
Oxygenates were the largest emission (33-55%), followed by furanic compounds (16-21%),
phenolics (6-12%) and aromatics (2-9%) for all fuel types except LPG. LPG emissions were
mainly alkanes, with a small emission of furanic species. These have previously been reported
to be produced in hydrocarbon flames (Johansson et al., 2016).
Figure 8A-B also show the amount of NMVOC which remained unidentified (black). On
average 94% of all NMVOCs emitted across all burns were quantified. Quantification was
greater than 90% for all sample types, except *Vachellia spp* due to several large unidentified
peaks (see the Supplementary Information S5). Mean quantification by fuel type was between
93-96 % for all other fuels, except LPG where quantification was > 99%.

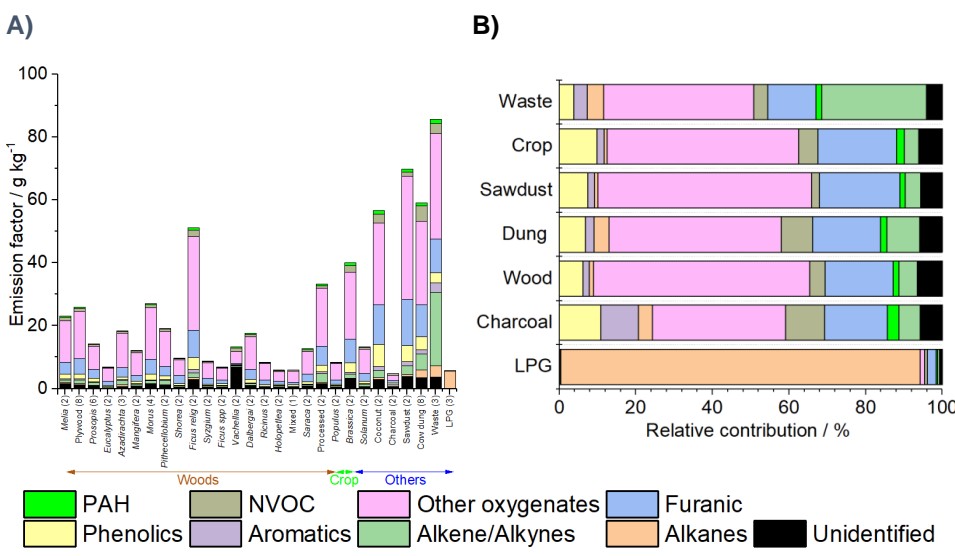

Figure 8. NMVOC emissions from burning sources in New Delhi, India grouped by functionality with
unidentified emissions given by the total NMVOC signal measured by the PTR-ToF-MS minus the
fraction quantified using DC-GC-FID, GCxGC-FID, GCxGC-ToF-MS and PTR-ToF-MS instruments
with (A) all fuel types and (B) mean values by type of fuel.



The emission factors measured in this study were compared to mean values measured in other
studies using PTR-ToF-MS (see Figure 9) for fuel wood, straw, peat and cow dung cake (Koss
et al., 2018); grasses, straws and peat (Stockwell et al., 2015) and forest fires (Simpson et al.,
2011; Müller et al., 2016; Liu et al., 2017). They were also compared to mean values calculated
from reviews for savannah, boreal forest, tropical forest, temperate forest, peatland, chaparral
and open cooking (Akagi et al., 2011) and savannah, tropical forest, temperate forest, boreal
forests, peat fires and biofuels without fuel wood (Andreae, 2019). Comparison was also made
to reviews for mean emission factors from just fuel woods from savannah, boreal forest,
tropical forest and temperate forest (Akagi et al., 2011; Andreae, 2019).
Figure 9A shows that emission factors measured in this study and those measured by Stockwell
et al. (2015), Koss et al. (2018), Muller et al. (2016) and Simpson et al. (2011) were generally
within a factor of 2-4. The differences in emission factors were likely due to differences in
composition between fuels collected from different locations. The emission factors measured
in this study were generally smaller than those reported in reviews by Akagi et al. (2011) and
Andreae (2019), despite the total NMVOC emission in this study being greater due to
measurement of a much wider range of NMVOCs. Emission factors for cow dung cake
measured in this study were closer to the 4:1 line, which showed that cow dung cake was
consistently more polluting per mass burnt than fuel wood.

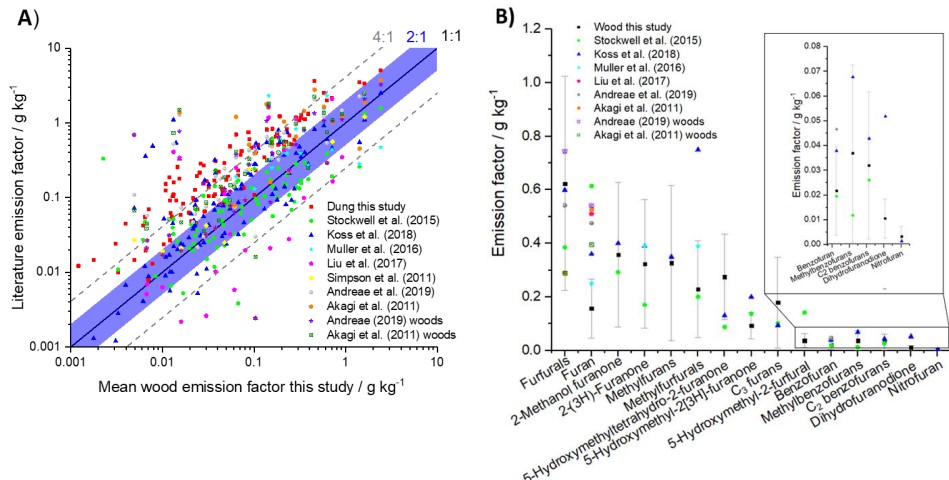

Figure 9. Emission factor comparison between this study and literature for (A) all species reported in
reviews and comparable studies and (B) furanic species from fuel woods. See text for discussion of fuel
types included in each study.





Figure 9B shows a comparison of emission factors for furanic species from fuel woods
compared with those from literature, which showed relatively good agreement within the
standard deviation observed from this study. A few notable exceptions were that the emission
factor for furfural measured by Muller et al. (2016) was considerably higher (2.3 g kg$^{-1}$) than
this study (0.7 g kg$^{-1}$), or previous studies, and not included in Figure 9B. The mean emission
factor for furan measured in this study was ~ factor 2 lower than other studies. Koss et al.
(2018) measured ~ factor 3 higher methyl furfural, ~ 9 higher 5-hydroxymethyl-2-furfural and
~ factor 3 higher dihydrofurandione and Stockwell et al. (2015) measured a higher emission
factor than this study of 5-hydroxymethyl-2-furfural by ~ factor of 4.
**3.5 Emission ratios**
The ratio of the mixing ratios of NMVOCs in the emitted gas can be a useful indicator of their
source(s) in ambient air. Ratios can be specific to sources and can allow one source to be
distinguished from another. The ratio of $i$-/$n$-pentane can be a useful indicator of whether
emissions are anthropogenic or from biomass burning, with a ratio 2.2-3.8 indicative of
vehicular emissions, 0.8-0.9 for natural gas drilling, 1.8-4.6 for evaporative fuel emissions and
< 1 from burning (Stewart et al., 2020b). Benzene/toluene ratios can also be useful and have
been reported from traffic exhaust to be around 0.3 (Hedberg et al., 2002).
$i$-/$n$-Pentane indicator ratios have been evaluated for fuel wood sources, propane/butane ratios
for LPG and benzene/toluene ratios for fuel wood and cow dung cake (see Figure 10). The
range of values for multiple different burns have been evaluated rather than just reporting mean
and median ratios. The median of $i$-/$n$-pentane ratios from biomass samples measured during
this study was ~ 0.7 (see Figure 10). The mean ratio was ~ 1.0, with an interquartile range
(IQR) ~ 0.5-1.5, which suggests caution is required when assigning burning sources based on
emission ratios due to considerable variability. Despite this, the ratio from solid fuel
combustion sources was often less than expected from petrol emissions. The mean ratio of
propane/butane from LPG burning was measured to be 3.1. The ratios of benzene/toluene
varied considerably between different sources and was measured for fuel wood combustion
(2.3), cow dung cake combustion (0.94), petrol liquid fuel (0.40) and diesel liquid fuel (0.20).
The range of benzene/toluene ratios for fuel wood was large, with an IQR of ~ 1.5- 2.8 and the
range within 1.5 IQR shown by the whiskers in Figure 10 from ~ 0.9-4.2. Despite the variability
of ratios from specific source types, the considerable range of benzene/toluene ratios could
potentially be a useful indicator of the origin of unaged (fresh) ambient emissions in New Delhi.
However, further study would be required to assess if these ratios were also true in the exhaust





of petrol and diesel vehicles in India, or just limited to fugitive emissions. These findings agree
well with literature which report mean benzene/toluene ratios of 1.4-5.0 from fuel wood and
0.3 from automotive emissions (Hedberg et al., 2002), indicating that on average biomass
burning releases a greater molar ratio of benzene than toluene when compared to automotive
emissions.

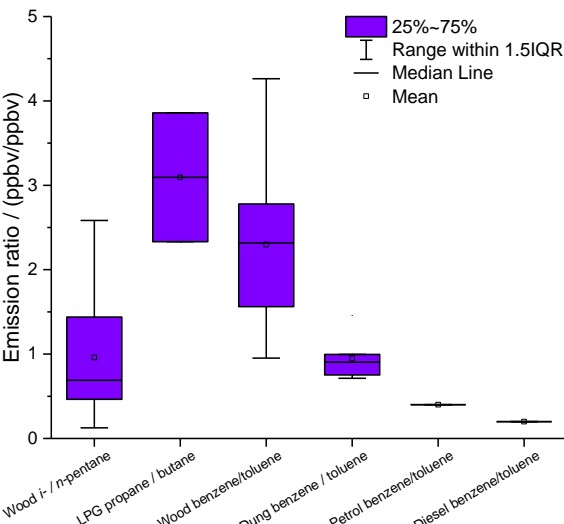

Figure 10. Summary of ratios of NMVOCs measured during this study from the burning of fuel wood,
LPG and cow dung cake and from the headspace of liquid petrol and diesel fuels collected in India. The
different mean and median values have been considered to evaluate the ratios at emission of specific
sources.

### 4. Conclusions

This study was based on comprehensive measurements of NMVOC emissions using a range of
detailed and complementary techniques across a large range of functionalities and volatilities.
It presented detailed burning emission factors for different NMVOCs from a range of fuels
used in New Delhi, India for residential combustion. This work allowed for a better
understanding of the impact of residential combustion on air quality and showed that fuel wood,
cow dung cake and municipal solid waste burning sources released significantly more
NMVOCs than LPG.
A range of areas where future studies are required to better improve and understand emissions
from burning have been highlighted:





1.      Better understanding of stove burn conditions on emissions

The impact of stove conditions on NMVOC emissions remains poorly understood. Experiments in this study were carried out using expert local judgment to attempt to ensure that laboratory conditions reflected real-world burning conditions. A range of stoves are used in India for combustion of local fuels, such as *chulha* and *angithi* stoves, and an evaluation of the impact of these on emissions and their relative use and spatial distribution requires further study.

2.      Better understanding of the effect of moisture content on modified combustion efficiency

Fuels in this study were collected and stored in a manner designed to be reflective of local practices to ensure that laboratory combustion conditions, and in turn emissions, reflected local burning practices. Future studies should conduct detailed compositional analysis of fuel types and moisture content prior to burning. These studies should also measure CO and $CO_2$ to allow an evaluation of the impact of modified combustion efficiency on emissions from different fuel types.

3.      Limited measurements of some fuel types

Few measurements were made from domestic, commercial and industrial waste, and the emission factors measured in this study were higher than those observed in previous studies. The effect of moisture content on waste burning has been suggested to impact emissions of particulate matter by around an order of magnitude (Jayarathne et al., 2018). Furthermore, only one LPG stove was used to evaluate emissions from this fuel source, with emissions likely to vary by the type of burner used. Future studies should also make more measurements from waste burning to better understand the effect of composition on emissions. Comprehensive measurements should also be made of emissions from combustion of a range of additional crop residues, as these are an important NMVOC source in India (Jain et al., 2014).

4.      Evaluation of the impact on $O_3$ and SOA production as well as the toxicity of emissions

Better understanding of the drivers of photochemical $O_3$ and SOA production from burning emissions is required. A large variety of high molecular weight species with likely low volatilities, such as phenolic and furanic compounds, were released from burning. These NMVOCs are expected to have a large influence on subsequent atmospheric chemistry, and a

detailed understanding of this chemistry is required to truly assess the impact of biomass
burning on air quality.
5.      Evaluation of the relative importance of fuel types to air quality in India
Detailed evaluation of fuel use across India is required to evaluate the relative impact of
emissions from fuel wood, municipal solid waste, cow dung cakes and LPG. The emission
factors measured for cow dung cake and municipal solid waste in this study were much higher
than for fuel wood and LPG and indicated that these sources are likely to contribute
significantly to poor air quality.
The comprehensive characterisation of emissions from fuel types in this study should be used
to produce spatially disaggregated local emission inventories to provide better inputs into
regional policy and global chemical transport models. This should allow a better understanding
of the key drivers of poor air quality in India and could allow meaningful mitigation strategies
to alleviate the poor air quality observed.
*Author contributions.* GJS made measurements with GC×GC-FID, combined and analysed
datasets and lead the writing of the manuscript. WJFA made measurements of NMVOCs by
PTR-ToF-MS, supported by CNH, LKS and NT. BSN made measurements with DC-GC,
supported by JRH. ARV assisted in running and organising of experiments. RA, AM, RJ, SA,
LY and SKS collected fuels, carried out burning experiments and measured gas volumes up
the flue. RED worked on GC×GC-FID method development. SSMY assisted with data
interpretation. EN, NM, RG, ARR and JDL worked on logistics and data interpretation. TKM
and JFH provided overall guidance with setup, conducting, running and interpreting
experiments. All authors contributed to the discussion, writing, and editing of the manuscript.
*Competing interests.* The authors declare that they have no conflict of interest.
*Acknowledgements.* This work was supported by the Newton-Bhabha fund administered by the
UK Natural Environment Research Council, through the DelhiFlux project of the Atmospheric
Pollution and Human Health in an Indian Megacity (APHH-India) programme. The authors
gratefully acknowledge the financial support provided by the UK Natural Environment
Research Council and the Earth System Science Organization, Ministry of Earth Sciences,
Government of India under the Indo-UK Joint Collaboration vide grant nos NE/P016502/1 and
MoES/16/19/2017/APHH (DelhiFlux) to conduct this research. The paper does not discuss
policy issues and the conclusions drawn in the paper are based on interpretation of results by



the authors and in no way reflect the viewpoint of the funding agencies. GJS and BSN
acknowledge the NERC SPHERES doctoral training programme for studentships. RA, AM,
RJ, SA, LY, SKS and TKM are thankful to Director, CSIR-National Physical Laboratory, New
Delhi for allowing to carry out this work. The authors thank the National Centre for
Atmospheric Science for providing the DC-GC-FID instrument. LKS acknowledges Physical
Research Laboratory (PRL), Ahmedabad, India for the support and permission to deploy PTR-
ToF-MS during the experimental campaign.





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
