# Peer review of "Emissions of non-methane volatile organic compounds from combustion of domestic fuels"

_Atmospheric Chemistry and Physics, 2020_

## Referee Comment (RC1) · Anonymous Referee #2 · 10 Nov 2020

In this paper, Stewart et al. present emission factors (EFs) for non-methane volatile organic compounds (NMVOC) produced during residential solid fuel combustion for heating and cooking in Delhi, India. Common cooking fuels were collected from across New Delhi and burned under controlled laboratory conditions. NMVOC emissions were measured using multiple gas chromatography based systems and a proton-transfer-reaction time-of-flight mass spectrometer. Species-specific and total measured NMVOC EFs are reported in the paper/supplement for each fuel type burned and the results are discussed in the context of similar laboratory studies of residential biomass burning emissions. The authors find that for most fuels oxygenated NMVOCs account for the largest proportion of the total NMVOC emissions. Additionally, they

report that the speciation and total measured NMVOC emissions vary widely between different fuel types, highlighting the need for a more complete understanding of residential biomass burning EFs. The study focuses on fuels that are specific to India and New Delhi, making their results relevant to local and regional chemical models. Emission factors from this study are also more broadly applicable to other regions where residential solid fuel combustion is used as the primary means of cooking and heating.

This clearly written manuscript addresses an underrepresented area of biomass burning emissions with a very comprehensive NMVOC EF dataset. I particularly appreciated the inclusion of LPG emissions to demonstrate its potential as a 'cleaner' alternative. I recommend this paper for publication after addressing the minor issues discussed below.

1) My main concern is the representativeness of the burning chamber used in this study to the common stoves used in residential settings. Although the chamber description is referenced, I feel that it is important for this study to include a more detailed description of the stove/combustion chamber itself along with how it was operated to replicate real-world conditions as the combustion effieciency is well known to influence NMVOC EFs.

2) There is very little discussion about the error associated with the reported EFs beyond that associated with each instrument, nor is the EF variability between repeated burning experiments of similar fuels included. For example, the mean EFs in the supplement and Table 2 should be associated with the fire-to-fire variance, such as the standard deviation of the burns. Similarly, what is the error associated with the stack flow-based method for determining EFs?

3) It is unclear whether CO and $CO_2$ were measured during the experiment, but if available their inclusion as EFs and MCEs for each burn would greatly help anchor this study in the context of NMVOC emission literature as the authors discuss in the conclusion.

Additional comments by line:

110 – Add missing word: due 'to' vehicular emissions.

198 – Repeated word 'given'.

414 – Should this be referencing S4? S2 appears to show how EFs were calculated. Additionally, S2, describing how EFs were calculated, is not referenced in the text and should be added. This also goes for the tables in supplement S4, they should be referenced in the text and would benefit from having the individual tabs labeled (Sx).

498 – Discussion of total emission factors would be more accurately discussed as total 'measured' emission factors as the techniques used in this study likely miss a portion of low volatility species, which could be lost in sample inlets and chromatography columns.

502-508 – It is unclear if you included the GC measurements in your total emission factors from this discussion. Is that the purpose of discussing proton affinities here? If so just state that alkanes and alkenes measured by the GC's were included in the total EF as appears to be described in S3.

509 – Should this be S3? Or maybe relevant to both S3 and S4 'EF g kg' tab?

512 – Include the mass of benzaldehyde.

544 – Stockwell et al. (2015), which the authors compare results to, define IVOC/SVOC as species with a molecular weight greater than toluene. Is there a reason the authors instead choose to define IVOCs as those with molecular weight greater benzaldehyde?

587 – Figure 9A – The different studies in the plot are very difficult to distinguish and it is unclear what the authors are trying to convey with it due to the potential comparison of unrelated EFs. For example, what does the inclusion of 'all species reported in review and comparable studies' include? Are wood emission factors from this study compared to garbage burning or peat EFs for Stockwell et al. (2016)? Are EFs from

western US wildfires (Liu et al., 2017) and southeastern US agricultural fires (Muller et al., 2016) relevant to this work? Further, EFs also vary between fuels due to differences in combustion efficiency (e.x. dung, peat, and trash will smolder more than wood) whereas this figure implies fuel type is the only difference. It would make sense to have this as a more direct comparison between related fuels (i.e. just literature fuel wood, cow dung, etc.).

650-657 – As mentioned, if available, this study would also greatly benefit from reporting CO, $CO_2$, and MCE values for each burn. As the authors state, this would allow their EFs to be evaluated based on the impacts of MCE. Additionally, reporting MCE would allow these results to be more accurately compared to other studies, while CO and $CO_2$ are themselves important inputs for climate models.
* * *

---

## Referee Comment (RC2) · Anonymous Referee #1 · 8 Dec 2020

In this work, the authors measured emission factors of volatile organic compounds (VOCs) emitted from combustion of a variety of fuels commonly used in India. This is an important topic for both atmospheric chemistry and human health, since domestic fuel combustion is associated with one of the leading causes of morbidity and mortality globally. The authors used a comprehensive suite of analytical techniques, and measured emission factors for a wide array of VOCs. In particular, the combination of PTR-MS and multiple GC techniques is highly complementary and provides detailed information about the emissions. In addition, the fuels studied are very commonly used in India and would provide important data and insights. From experimental design to data analysis and interpretation, this work is of the highest quality and potential impact.

[Figure]

I recommend publication in ACP, and my suggestions and comments here are minor and for reference only.

One of the major strengths of this study is its direct relevance. The authors stated that they used "expert local judgement to ensure conditions replicated real world burning conditions". It is unclear what that means from a technical standpoint. Precisely what variables are replicated to reflect local practices? (e.g. fuel types, forms of fuel, humidity etc.) How should future studies replicate the results presented here?

In multiple instances, the authors noted that PTR-MS measured higher amounts than GC techniques for the same compounds, and attributed to "unidentified isomers". I am curious to learn more about this issue. If a particular PTR-MS m/z is assigned to a compound that has multiple isomers, and in GC there is an associated peak (which represents one of the isomers), shouldn't the comparison be made between PTR-MS m/z and the sum of all isomers measured by GC (i.e. sum of multiple peaks)? If the "unidentified isomers" are not observed by the GC, that would imply these "unidentified isomers" are chemically different from the proposed compound, and therefore PTR-MS is actually misidentifying these isomers.

It is not surprising that cow dung cake and municipal solid waste had the highest emission factor, but what is the typical quantity burned? I imagine the fuel wood would be much more commonly used. It might be useful to clarify whether with the high emission factors of cow dung cake and municipal solid waste translate to higher contributions of VOCs.

Figure 3 shows an interesting trend: there seems to be two rather distinct phases of burning in A and B. Was this typical of all burns? If so, why is that the case, and what do these phases represent?

Minor comments:

Line 37: "400 Tg yrˆ-1" and "annually" in line 38 are redundant

Line 110: due to

Line 131: unclear what 6000-7000 kt yrˆ-1 is referring to. Is it total VOCs?

Line 162: range of

Line 181: does the quartz filter potentially remove gas phase species such as IVOCs?

Figure 1: I understand that the x-axis is referring to longitude, but I initially mistook it as ion.

Line 306: Since this is the results section, the title "Chromatography" is not very helpful. I suggest a more descriptive title.

Figure 2: Are the units of the color scale arbitrary? Are each of the samples obtained with the same volume of air sampled? If so, it might be useful to clarify and emphasize that, because if the color scales and the air volume sampled were the same, then solid waste and cow dung are indeed emitting more VOCs.

Figure 3: it is not directly obvious to me that from Region A to C the average m/z is increasing. Perhaps show the median mass, or overlay these diagrams, or stack them vertically with a common x-axis?

Line 380: How is ACES an abbreviation of broadband cavity-enhanced spectroscopy?

Line 383: 1 +/- 30% can be misleading. I suggest 1+/- 0.3

Line 387: "These previous comparisons underline the challenges faced with quantitative NMVOC measurements..." this sentence seems to contradict the previous sentences. It seems that correlation coefficients are generally >0.8 in the literature, which is the same as what was obtained in this study. It seems to be this level of consistency is to be expected. (Perhaps that's what the authors mean?)

Figure 7 may be too detailed and many of the labels are far too small to see. I struggle to see the message conveyed by these figures. I suggest showing figures that support

[Figure]

the discussion in 3.4 and minimize information overload. Line 528: "however" might not be the best conjunction. "But" is more grammatically correct.

Line 547: Since IVOCs are being reported, what is the typical PM concentration? Are the PM concentrations high enough for IVOC to partition into the particle phase?
* * *

---

## Author Comment (AC1) · 17 Dec 2020

We would like to thank the reviewers for their positive and constructive reviews of this paper. We address the specific points of each reviewer below. Reviewer comments in blue, author response in black, text added or amended in paper in purple.

Review 1

In this paper, Stewart et al. present emission factors (EFs) for non-methane volatile organic compounds (NMVOC) produced during residential solid fuel combustion for heating and cooking in Delhi, India. Common cooking fuels were collected from across New Delhi and burned under controlled laboratory conditions. NMVOC emissions were measured using multiple gas chromatography-based systems and a proton transfer-reaction time-of-flight mass spectrometer. Species-specific and total measured NMVOC EFs are reported in the paper/supplement for each fuel type burned and the results are discussed in the context of similar laboratory studies of residential biomass burning emissions. The authors find that for most fuels oxygenated NMVOCs account for the largest proportion of the total NMVOC emissions. Additionally, they report that the speciation and total measured NMVOC emissions vary widely between different fuel types, highlighting the need for a more complete understanding of residential biomass burning EFs. The study focuses on fuels that are specific to India and New Delhi, making their results relevant to local and regional chemical models. Emission factors from this study are also more broadly applicable to other regions where residential solid fuel combustion is used as the primary means of cooking and heating. This clearly written manuscript addresses an underrepresented area of biomass burning emissions with a very comprehensive NMVOC EF dataset. I particularly appreciated the inclusion of LPG emissions to demonstrate its potential as a 'cleaner' alternative. I recommend this paper for publication after addressing the minor issues discussed below.

1) My main concern is the representativeness of the burning chamber used in this study to the common stoves used in residential settings. Although the chamber description is referenced, I feel that it is important for this study to include a more detailed description of the stove/combustion chamber itself along with how it was operated to replicate real-world conditions as the combustion efficiency is well known to influence NMVOC EFs.

The chamber was based on a previously published design of Venkataraman and Rao, (2001). The chamber was designed to simulate the convective nature of biomass combustion, so it

was important to ensure that the processes studied here of emissions entrainment into the hood were also convection driven so that they did not exert a draft which altered combustion conditions. The dilution setup employed here was optimised to give dilution ratios of 40-60, which allowed cooling of gases to around 2-3 °C above ambient temperature at the top of the flue.

This chamber has been previously tested and optimised to ensure that conditions replicate those of a natural draft during combustion. The burn rate has been previously evaluated using extraction rates of 0.01-0.03 $m^3 s^{-1}$ and stove-hood distances of 0.35-0.65 m. Larger extraction rates and stove-hood distances less than 0.45 m enhanced burn rates above the natural burn rate. Stove-hood distances above 0.65 m resulted in emissions not being captured by the hood. The optimum conditions were used of 0.45 m between sample and hood with a flow rate of ~ 0.022 $m^3 s^{-1}$.

We now include additional details of the chamber used in the main text and a detailed schematic of the chamber is given in the supplementary information. We also include a video abstract which shows the ignition of a sample during this study to show the conditions this study was designed to replicate. Whilst several different stove types can be used in India, this study was most like a traditional fire. The main text has now been changed.

Fuels were burnt at the CSIR-National Physical Laboratory (NPL), New Delhi, under controlled conditions utilizing a combustion chamber based on the design of Venkataraman and Rao, (2001). Several previous studies have been based on this chamber design (Venkataraman and Rao, 2001; Venkataraman et al., 2002; Saud et al., 2011; Saud et al., 2012; Singh et al., 2013), which was designed to simulate the convection-driven conditions of real-world combustion and is displayed in the Supplementary Information S1. The burn-cycle used in this study was adapted from the VITA water-boiling test, which is designed to simulate emissions from cooking, using expert local judgement to ensure conditions replicated real-world burning conditions. The cycle included emissions from both low- and high-temperature burning conditions, as these are encountered in real cooking practice and should give a more reflective NMVOC emission factor.

Fuel (200 g) was placed 45 cm from the top of the hood and rapidly heated to spontaneous ignition, with emissions convectively driven into a hood and up a flue to allow enough dilution,

cooling and residence time to achieve the quenching typically observed in indoor environments. These conditions have been previously optimised to ensure that emissions entrainment into the hood did not exert a draft which altered combustion conditions.

This schematic of the burning chamber has been added to the supplementary information.

[Figure]

Figure 1. Schematic of combustion chamber used for experiments.

2) There is very little discussion about the error associated with the reported EFs beyond that associated with each instrument, nor is the EF variability between repeated burning experiments of similar fuels included. For example, the mean EFs in the supplement and Table 2 should be associated with the fire-to-fire variance, such as the standard deviation of the burns.

The standard deviation of measured NMVOC emission factors by sample type are presented in Table 1 in the main text. We also now look at the EF variability of similar fuels in detail within the main text, as detailed below.

Figure 2A shows the distribution of total measured NMVOC emission factors for fuel wood, cow dung cake, crop residues and MSW. Boxplots show the mean, median, interquartile range and range within 1.5IQR. The solid circles display the spread of measured emission factors by fuel type. The zoomed green region given in Figure 2B specifically focuses on the variability in emission factors of individual species of fuel wood, which has been explored in detail due to the large number of samples. Repeat samples collected from the same location are shaded in grey. For fuel wood, measured NMVOC emission factors varied by over a factor of 20 between 4.3-96.7 g kg$^{-1}$. The NMVOC emission factors showed a right skewed distribution with a median of 11.7 g kg$^{-1}$, mean of 18.7 g kg$^{-1}$ and an interquartile range of 15.3 g kg$^{-1}$. For repeat measurements of identical species of fuel wood collected at the same location, except for *Ficus religiosa*, measured emission factors from repeat experiments varied over a much smaller range, by up to a factor of 2.3. Variation between emissions from these samples were likely due to different moisture contents of actual samples measured and the specific combustion conditions of individual burns. Despite the samples for *Holopetlea spp* and *Eucalyptus spp* coming from different locations, emission factors for these samples were quite reproducible and only varied by a factor of 1.2-1.5. For remaining identical species of fuel wood collected from different locations, emission factors varied over a much larger range by factors of ~ 2-9.

For the crop residue species studied here, NMVOC emissions were right skewed with a with a median of 29.5 g kg$^{-1}$ which was less than the mean of 37.9 g kg$^{-1}$ and varied from 8.9-73.8 g kg$^{-1}$ with an interquartile range of 53.9 g kg$^{-1}$. *Cocos nucifera* and *Solanum melongena* were repeat measurements of fuel collected from the same location and varied by factors of 1.8-2. NMVOC emissions from *Brassica spp* fuel, which was collected from different locations, varied by a factor of ~ 8. Cow dung cake and MSW samples were all collected from different locations and varied by up to factors of up to 2.4 and 2.1, respectively.

[Figure]

Figure 2. Variability in NMVOC emission factor by fuel type. A) = Range of emission factors measured for fuel wood, cow dung cake, crop residue and municipal solid waste samples with box plots showing the mean, median, interquartile range, range within 1.5IQR and solid circles showing the spread of measured emission factors by fuel type. B) = Zoomed green region displaying range of NMVOC emission factors measured for individual species of fuel wood with grey shaded region indicating repeat samples from the same sample collection location and diamonds indicating the measured NMVOC emission factors.

Similarly, what is the error associated with the stack flow-based method for determining EFs?

Venkataraman and Rao, (2001) studied the stack-flow based method for determining EFs. As part of this study the reproducibility of dilutions from repeat fires was examined, with Table 1 giving the results at 1 σ of 4 repeat measurements.

Table 1. Repeatability of dilution ratios using stack flow-based method, taken from Venkataraman and Rao, (2001).

| Sample | Dilution ratio |
|---|---|
| Wood | 57 ± 6 |
| | 47 ± 7 |
| | 46 ± 8 |
| | 53 ± 3 |
| | |
| Biofuel briquette | 40 ± 3 |
| | 42 ± 7 |
| Dung cake | 56 ± 13 |
| | 42 ± 9 |
| | 43 ± 6 |
| | 60 ± 7 |

3) It is unclear whether CO and $CO_2$ were measured during the experiment, but if available their inclusion as EFs and MCEs for each burn would greatly help anchor this study in the context of NMVOC emission literature as the authors discuss in the conclusion.

CO and $CO_2$ were originally intended to be measured during this study, however, due to a technical failure there was a lack of sufficiently reliable data to be of use to include here. The authors acknowledge the importance of emission factors to CO and $CO_2$ as well as the influence of modified combustion efficiency to NMVOC emissions from burning studies, but unfortunately cannot include this here.

Additional comments by line:

110 – Add missing word: due 'to' vehicular emissions.

This is now corrected.

198 – Repeated word 'given'. 414 – Should this be referencing S4?

This is now corrected.

S2 appears to show how EFs were calculated. Additionally, S2, describing how EFs were calculated, is not referenced in the text and should be added. This also goes for the tables in supplement S4, they should be referenced in the text and would benefit from having the individual tabs labeled (Sx).

The text has been changed to read, where S2 is now S3 due to reordering.

**Error! Reference source not found.** shows a detailed breakdown of the mean NMVOC emission factors by fuel type measured for all 76 burns (see the Supplementary Information S3 for values). Emission factors have been determined by calculating the mean NMVOC concentrations up the flue over a 30-minute period, in line with the GC sample time, with any small emissions after this sample window not included. This has been related to the total volume of air convectively drawn up the flue and the mass of fuel burnt (see the Supplementary Information S4 for details).

The tabs in the supplementary table are now labelled too.

498 – Discussion of total emission factors would be more accurately discussed as total 'measured' emission factors as the techniques used in this study likely miss a portion of low volatility species, which could be lost in sample inlets and chromatography columns.

We now use the terminology total measured emission factor throughout.

502-508 – It is unclear if you included the GC measurements in your total emission factors from this discussion. Is that the purpose of discussing proton affinities here? If so just state that alkanes and alkenes measured by the GC's were included in the total EF as appears to be described in S3.

This is now stated.

509 – Should this be S3? Or maybe relevant to both S3 and S4 'EF g kg' tab? 512 – Include the mass of benzaldehyde.

This is now clarified in the text.

544 – Stockwell et al. (2015), which the authors compare results to, define IVOC/SVOC as species with a molecular weight greater than toluene. Is there a reason the authors instead choose to define IVOCs as those with molecular weight greater benzaldehyde?

This is a good question, as saturation vapour concentration pressures depend on both mass and functionality and it is therefore difficult to define based on a particular mass. IVOCs are defined as having effective saturation concentration, $C^*$ =300-3×10$^6$ µg m$^{-3}$. We calculated these for NMVOCs in our mass spectra following the approach given in Lu et al, (2018).

The estimated $C^*$ for toluene was ~ 1.4×10$^8$ µg m$^{-3}$ and benzaldehyde ~ 7×10$^6$ µg m$^{-3}$. For this reason, we based the IVOC boundary on benzaldehyde and not toluene. The total amount of IVOCs presented from this study would therefore be a more conservative estimate of total measured IVOC compared to Stockwell et al. (2016). The reason for this approach has been emphasised in the text, and attention also brought to this approach being approximate. The text has been changed to read:

IVOCs are defined as having effective saturation concentration, $C^*$, =300-3×10$^6$ µg m$^{-3}$ (Donahue et al., 2012). The $C^*$ of several species was estimated using a previously established

approach (Lu et al., 2018), with the IVOC boundary defined in this study at benzaldehyde ($m$ = 106.12) for which $C*$ was ~ $7 \times 10^6$ µg m$^{-3}$. **Error! Reference source not found.** also shows an approximation for the mean amount of IVOCs released by fuel type. This approach was approximate as vapour pressures depend on both mass and functionality. The fuels tested in this study showed that mean emissions of IVOC species represented approximately 18 – 27% of total measured emissions from all fuel types other than LPG. This demonstrated that biomass burning is potentially a large global source of IVOCs. Further studies are required to better understand the contribution of IVOC emissions from biomass burning to SOA formation.

587 – Figure 9A – The different studies in the plot are very difficult to distinguish and it is unclear what the authors are trying to convey with it due to the potential comparison of unrelated EFs. For example, what does the inclusion of 'all species reported in review and comparable studies' include? Are wood emission factors from this study compared to garbage burning or peat EFs for Stockwell et al. (2016)? Are EFs from western US wildfires (Liu et al., 2017) and south eastern US agricultural fires (Muller et al., 2016) relevant to this work? Further, EFs also vary between fuels due to differences in combustion efficiency (e.x. dung, peat, and trash will smoulder more than wood) whereas this figure implies fuel type is the only difference. It would make sense to have this as a more direct comparison between related fuels (i.e. just literature fuel wood, cow dung, etc.).

This is a good point. We made the comparison to just woods from Akagi et al. (2011), Andreae (2019) and Koss et al (2018). Generally, the emission factors measured were larger from these studies than measured in our study. We also compared emission factors from waste burning in Stockwell et al. (2016) to our study. Despite this, the comparison was not particularly interesting. When we plotted these different studies, the other studies often had higher emission factors than our study for many data points, but there was considerable scatter in the data points both above and below the 1:1 line. This meant that including these additional plots in the main body of the text was not particularly beneficial, so we removed this section from the paper.

650-657 – As mentioned, if available, this study would also greatly benefit from reporting CO, CO2, and MCE values for each burn. As the authors state, this would allow their EFs to be evaluated based on the impacts of MCE. Additionally, reporting MCE would allow these

results to be more accurately compared to other studies, while CO and CO2 are themselves important inputs for climate models.

This is covered as part of the response to major comment 3.

In this work, the authors measured emission factors of volatile organic compounds (VOCs) emitted from combustion of a variety of fuels commonly used in India. This is an important topic for both atmospheric chemistry and human health, since domestic fuel combustion is associated with one of the leading causes of morbidity and mortality globally. The authors used a comprehensive suite of analytical techniques, and measured emission factors for a wide array of VOCs. In particular, the combination of PTR-MS and multiple GC techniques is highly complementary and provides detailed information about the emissions. In addition, the fuels studied are very commonly used in India and would provide important data and insights. From experimental design to data analysis and interpretation, this work is of the highest quality and potential impact.

We thank the reviewer for such a positive review and for highlighting the strengths of this work.

I recommend publication in ACP, and my suggestions and comments here are minor and for reference only. One of the major strengths of this study is its direct relevance. The authors stated that they used "expert local judgement to ensure conditions replicated real world burning conditions". It is unclear what that means from a technical standpoint. Precisely what variables are replicated to reflect local practices? (e.g. fuel types, forms of fuel, humidity etc.) How should future studies replicate the results presented here?

This has been partly covered by the response to main comment 1 from reviewer comment 1 through the additional discussion on chamber setup and design. Fuels were collected from residents of Delhi state from the same areas they collected their fuels to be burnt. This was designed so that the fuels which were burnt were identical to those burnt under real circumstances. The fuels were handled and stored as locals would to ensure that the moisture content of samples was like those being burnt for residential energy requirements. The

combustion chamber used has been previously studied to ensure the combustion conditions were convective and that neither the hood nor any fans nor pumps within the setup created a draft which altered combustion conditions and in turn NMVOC emissions.

Similar studies would follow the experimental design given in Venkataraman and Rao, (2001), which is now also given in the Supplementary Information. The fuel samples were collected from a detailed study, which in summary split the state of Delhi into 66 5×5 km grids and conducted fuel usage surveys at almost 700 locations of over 6000 households. The collected fuels were designed to reflect the results of this survey. The results of this survey will be presented in a separate publication (Mondal et al., 2021) led by the Indian co-authors in this study.

In multiple instances, the authors noted that PTR-MS measured higher amounts than GC techniques for the same compounds, and attributed to "unidentified isomers". I am curious to learn more about this issue. If a particular PTR-MS m/z is assigned to a compound that has multiple isomers, and in GC there is an associated peak (which represents one of the isomers), shouldn't the comparison be made between PTR-MS m/z and the sum of all isomers measured by GC (i.e. sum of multiple peaks)? If the "unidentified isomers" are not observed by the GC, that would imply these "unidentified isomers" are chemically different from the proposed compound, and therefore PTR-MS is actually misidentifying these isomers.

This is indeed one of the limitations of measurements with PTR-ToF-MS. Figure 6 gives a comparison of some aromatic compounds measured by the PTR-ToF-MS and both GC instruments. The PTR-ToF-MS instrument gives a signal at a particular mass, for which we have assigned the most probable identity. If we take the example of benzene, other potential $C_6H_6$ compounds include benzvalene, bicyclopropenyl, fulvene, prismane, 3-Methylidenepent-1-en-4-yne, Hexadiyne, 1,3-Hexadiyne, 1,4-Hexadiyne, 1,5-Hexadiyne, 2,4-Hexadiyne, Hexadienyne, 1,2-Hexadien-4-yne, 1,2-Hexadien-5-yne, 1,3-Hexadien-5-yne, 1,5-Hexadien-3-yne or 2,3-Hexadien-5-yne. All these other $C_6H_6$ compounds would be indistinguishable from benzene.

The GC instruments only have flame ionisation detectors, and so we are only able to calibrate peaks for which the identity is known through the retention time of a known standard compound. Peaks for other $C_6H_6$ compounds may be present in our chromatograms, but we

are not able to identify which these are. As a result, this comparison is currently benzene on the GC×GC-FID, benzene and coeluting peaks with the same retention time on the DC-GC-FID and all $C_6H_6$ compounds on the PTR-ToF-MS. The purpose of this comparison was therefore to show that the instruments were all measuring similar concentrations, but there may be some additional undistinguishable structural isomers measured on the PTR-ToF-MS instrument.

As a further complication, NMVOCs on the PTR-ToF-MS were calibrated with the rate constant for the reaction of the hydronium ion with the NMVOC of the most likely compound. Where multiple isomers were present, this may lead to slight mis quantification.

It is not surprising that cow dung cake and municipal solid waste had the highest emission factor, but what is the typical quantity burned? I imagine the fuel wood would be much more commonly used. It might be useful to clarify whether with the high emission factors of cow dung cake and municipal solid waste translate to higher contributions of VOCs.

This is a great comment and something we have prepared a further study on which is currently awaiting submission. It is also difficult to put a firm number on, as there are considerable uncertainties in fuel consumption estimates for India and many estimates are quite out of date. We have added the following text to help clarify this.

Considerable uncertainties exist in consumption estimates for fuels such as cow dung cake and municipal solid waste in India. A previous study estimated that in 1985 in India fuel wood consumption was 220 Tg and cow dung cake consumption 93 Tg (Yevich and Logan, 2003). A different study made an India-wide estimate for 2000 which estimated fuel wood consumption to be 281 (192-409) Tg and cow dung cake consumption to be 62 (35-128) Tg (Habib et al., 2004). A more recent study estimated fuel wood usage at 256 Tg and cow dung cake consumption at 106 Tg for 2007 (Singh et al., 2013). Estimates of the amount of municipal solid waste burnt in India are even fewer than for cow dung cake consumption. Two previous studies have estimated that 81.4 Tg of municipal solid waste was burnt in India in 2010 (Wiedinmyer et al., 2014) and that 68 (45-105) Tg was burnt in 2015 (Sharma et al., 2019). The mean emission factors for cow dung cake and municipal solid waste combustion were considerably larger than for fuel wood and highlight that at an India-wide level these may represent a significant NMVOC source.

Figure 3 shows an interesting trend: there seems to be two rather distinct phases of burning in A and B. Was this typical of all burns? If so, why is that the case, and what do these phases represent?

The phases may represent flaming and smouldering emissions. Sekimoto et al. (2018) showed that during lower temperature pyrolysis there were greater emissions of heavier molecular weight phenolic and furanic components. It may also be due to the off gassing of IVOC species from the quartz filter or the tubing used in this laboratory setup, which becomes more pronounced when the concentration of other gases is lower once the burning experiment has almost finished.

We are cautious to offer too much explanation of the difference in phases of the burn because the method used to measure total gas flow up the flue was used to give an integrated picture of the whole burn, and not used for a time-resolved measurement. Whilst this gives a quantitative measurement of the total volume of air sampled up the flue over the entire burn, it may slightly misrepresent the phases in Figure 3 where the average gas velocity over the entire burn has been used to calculate the concentration. Certain regions of the burn may be more (or less) pronounced here, which are not accounted for with this approach.

Minor comments: Line 37: "400 Tg yrˆ-1" and "annually" in line 38 are redundant

This has now been corrected.

Line 110: due to Line 131: unclear what 6000-7000 kt yrˆ-1 is referring to. Is it total VOCs?

This is a reference to total VOC emissions from burning in India. The text has been changed to read

India-specific inventories which include residential burning indicate a considerable emission source of total NMVOCs of around 6000-7000 kt yr$^{-1}$

Line 162: range of Line 181: does the quartz filter potentially remove gas phase species such as IVOCs?

This is a good comment, and it may and is therefore one of the limitations of this study. Between each sample there was a considerable amount of particulate matter collected onto these filters and if we did not change the filter between samples it increased the background

on the PTR-ToF-MS instrument. This may indicate some off gassing of species which may have partitioned to the aerosol phase. We would not want to run these instruments without a filter to remove particulate matter but feel that this is an important limitation, so we now acknowledge it in the text later with the discussion section about the proportion of IVOC species. It may also explain the larger mass fraction of IVOC species in region C of Figure 3 as this could be species off gassing from chamber walls or this filter, which enhances the proportion of IVOC species.

In addition, this may represent an underestimate because the quartz filter placed on the sample line may remove IVOC species which have partitioned to the aerosol phase due to the high aerosol concentrations present during source testing.

Figure 1: I understand that the x-axis is referring to longitude, but I initially mistook it as ion.

Thank you for highlighting this, we have now capitalised these axes so that they are clearer.

[Figure]

Figure 3. Locations across New Delhi used for the local surveys into fuel use and collection of representative biomass fuels. Map tiles by Stamen Design. Data by © OpenStreetMap contributors 2020. Distributed under a Creative Commons BY-SA License.

 Since this is the results section, the title "Chromatography" is not very helpful. I suggest a more descriptive title.

Thank you for this suggestion, we have changed this to read "Comparison of chromatograms obtained from combustion of different fuel types".

Figure 2: Are the units of the color scale arbitrary? Are each of the samples obtained with the same volume of air sampled? If so, it might be useful to clarify and emphasize that, because if the color scales and the air volume sampled were the same, then solid waste and cow dung are indeed emitting more VOCs.

Thank you for highlighting this. We have chosen this contrast scale between 0 and 25 as it allows a nice visualisation and comparison of the NMVOCs present. All the chromatograms are at the same level of contrast and samples were collected with the same sample volumes. We added to the caption on figure 2 Samples A-D were collected with the same sample collection parameters and the chromatograms are set at the same contrast level to allow direct comparison between different fuel types.

Figure 3: it is not directly obvious to me that from Region A to C the average m/z is increasing. Perhaps show the median mass, or overlay these diagrams, or stack them vertically with a common x-axis?

Thank you for this suggestion, we have now stacked the binned mass spectra of regions A and B vertically to better emphasise this change. We have removed the section from 500-700 seconds at this was showing a similar trend as 200-300 seconds.

[Figure]

Figure 4. PTR-ToF-MS concentration-time series during the first 30 minutes of a cow dung cake burn coloured by functionality with regions A and B displaying mass spectra placed into *m/z* bins of 10 Th. Fuel collected from Pitam Pura, New Delhi.

Line 380: How is ACES an abbreviation of broadband cavity-enhanced spectroscopy?

Thank you for pointing this out, we have changed it to read airborne cavity-enhanced spectroscopy (ACES).

Line 383: 1 +/- 30% can be misleading. I suggest 1+/- 0.3 Line 387:

Thank you for the suggestion, we have changed this to read 1 ± 0.3%.

"These previous comparisons underline the challenges faced with quantitative NMVOC measurements. . ." this sentence seems to contradict the previous sentences. It seems that correlation coefficients are generally >0.8 in the literature, which is the same as what was obtained in this study. It seems to be this level of consistency is to be expected. (Perhaps that's what the authors mean?)

Thank you for this suggestion, we have changed this section to read These previous comparisons indicate the level of consistency expected with instrument comparisons of quantitative NMVOC measurements from burning experiments.

Figure 7 may be too detailed and many of the labels are far too small to see. I struggle to see the message conveyed by these figures. I suggest showing figures that support the discussion in 3.4 and minimize information overload.

Thank you for this comment, we agree that this figure is too small. This was partly due to the need for portrait figures in ACPD. We have changed it so that it covers 2 full pages in landscape format. We have also added some additional shaded boxes to help highlight which areas correspond to specific fuel types.

We would like to keep the information presented, to allow readers to quickly glance by class of NMVOC to see if it is important to their study or interest. We feel presenting this graphically is easier for users to determine if classes of VOC are important for particular fuel types than looking through the table in the supplementary information which contains 76 rows of different burns and almost 200 columns of NMVOCs.

[Figure]

Figure 7. Measured emission factors grouped by functionality.

[Figure]

Figure 7 continued.

Line 528: "however" might not be the best conjunction. "But" is more grammatically correct.

Thank you for this suggestion, we have changed this to but.

Line 547: Since IVOCs are being reported, what is the typical PM concentration? Are the PM concentrations high enough for IVOC to partition into the particle phase?

Thank you for this great question. Traditional source studies make separate gas-phase and particle-phase measurements of organic emissions, and so if the study is only of gas-phase emissions and IVOCs partition to the particle phase because of the unrealistically high particulate matter concentrations during source testing then these are not accounted for and therefore underestimated in the emission factor measurement. The gas-phase emission factors presented in this study may therefore represent an underestimate. We now acknowledge this in the text with the previous comment about discussion of the quartz filter.

We attempt to help overcome this as part of a further study, which is currently in review with a different journal, where we map the emissions from the DC-GC-FID, GC×GC-FID, PTR-ToF-MS and SPE/PTFE-GC×GC-ToF-MS analyses onto a volatility basis dataset to evaluate organic emissions across the entire volatility range and remove this traditional gas/aerosol phase divide when analysing sources at the point of emission.

**References**

Donahue, N. M., Kroll, J. H., Pandis, S. N., and Robinson, A. L.: A two-dimensional volatility basis set – Part 2: Diagnostics of organic-aerosol evolution, Atmos. Chem. Phys., 12, 615-634, 10.5194/acp-12-615-2012, 2012.

Habib, G., Venkataraman, C., Shrivastava, M., Banerjee, R., Stehr, J. W., and Dickerson, R. R.: New methodology for estimating biofuel consumption for cooking: Atmospheric emissions of black carbon and sulfur dioxide from India, Global Biogeochemical Cycles, 18, 10.1029/2003GB002157, 2004.

Lu, Q., Zhao, Y., and Robinson, A. L.: Comprehensive organic emission profiles for gasoline, diesel, and gas-turbine engines including intermediate and semi-volatile organic compound emissions, Atmos. Chem. Phys., 18, 17637-17654, 10.5194/acp-18-17637-2018, 2018.

Saud, T., Mandal, T. K., Gadi, R., Singh, D. P., Sharma, S. K., Saxena, M., and Mukherjee, A.: Emission estimates of particulate matter (PM) and trace gases ($SO_2$, NO and $NO_2$) from biomass fuels used in rural sector of Indo-Gangetic Plain, India, Atmospheric Environment, 45, 5913-5923, https://doi.org/10.1016/j.atmosenv.2011.06.031, 2011.

Saud, T., Gautam, R., Mandal, T. K., Gadi, R., Singh, D. P., Sharma, S. K., Dahiya, M., and Saxena, M.: Emission estimates of organic and elemental carbon from household biomass fuel used over the Indo-Gangetic Plain (IGP), India, Atmospheric Environment, 61, 212-220, https://doi.org/10.1016/j.atmosenv.2012.07.030, 2012.

Sekimoto, K., Koss, A. R., Gilman, J. B., Selimovic, V., Coggon, M. M., Zarzana, K. J., Yuan, B., Lerner, B. M., Brown, S. S., Warneke, C., Yokelson, R. J., Roberts, J. M., and de Gouw, J.: High- and low-temperature pyrolysis profiles describe volatile organic compound emissions from western US wildfire fuels, Atmos. Chem. Phys., 18, 9263-9281, 10.5194/acp-18-9263-2018, 2018.

Sharma, G., Sinha, B., Pallavi, Hakkim, H., Chandra, B. P., Kumar, A., and Sinha, V.: Gridded Emissions of CO, $NO_x$, $SO_2$, $CO_2$, $NH_3$, HCl, $CH_4$, $PM_{2.5}$, $PM_{10}$, BC, and NMVOC from Open Municipal Waste Burning in India, Environmental Science & Technology, 53, 4765-4774, 10.1021/acs.est.8b07076, 2019.

Singh, D. P., Gadi, R., Mandal, T. K., Saud, T., Saxena, M., and Sharma, S. K.: Emissions estimates of PAH from biomass fuels used in rural sector of Indo-Gangetic Plains of India, Atmospheric Environment, 68, 120-126, https://doi.org/10.1016/j.atmosenv.2012.11.042, 2013.

Venkataraman, C., and Rao, G. U. M.: Emission Factors of Carbon Monoxide and Size-Resolved Aerosols from Biofuel Combustion, Environmental Science & Technology, 35, 2100-2107, 10.1021/es001603d, 2001.

Venkataraman, C., Negi, G., Brata Sardar, S., and Rastogi, R.: Size distributions of polycyclic aromatic hydrocarbons in aerosol emissions from biofuel combustion, Journal of Aerosol Science, 33, 503-518, https://doi.org/10.1016/S0021-8502(01)00185-9, 2002.

Wiedinmyer, C., Yokelson, R. J., and Gullett, B. K.: Global Emissions of Trace Gases, Particulate Matter, and Hazardous Air Pollutants from Open Burning of Domestic Waste, Environmental Science & Technology, 48, 9523-9530, 10.1021/es502250z, 2014.

Yevich, R., and Logan, J. A.: An assessment of biofuel use and burning of agricultural waste in the developing world, Global Biogeochemical Cycles, 17, 10.1029/2002GB001952, 2003.